# FRAGMENT-AUGMENTED DIFFUSION FOR MOLECULAR CONFORMATION GENERATION

## ABSTRACT

Molecular conformer generation is a fundamental challenge in computational chemistry, particularly for large and complex molecules. In this work, we propose a novel approach called Fragment-Augmented Diffusion (FADiff), which integrates molecular fragmentations into diffusion models as a data augmentation strategy to enhance molecular conformation generation. By decomposing molecules into smaller, manageable fragments for the purpose of data augmentation, FADiff enhances the diffusion generation process, effectively capturing local structural variations while preserving the integrity of the entire molecule. Extensive experiments across multiple datasets demonstrate that FADiff consistently outperforms state-of-the-art methods, particularly in data-scarce scenarios, where the fragment-based augmentation approach significantly enhances model performance. We also provide a comprehensive analysis of different fragmentation rules and their impact on model performance, and theoretically validate FADiff's effectiveness in improving generalization. Overall, FADiff advances molecular conformation generation by enhancing the exploration of conformational space, offering a powerful tool for computational chemistry. The code is available at https://anonymous.4open.science/r/fragaug-5960/.

## 1 INTRODUCTION

The generation of 3D molecular conformations is a cornerstone in computational chemistry, crucial for understanding molecular properties and interactions. Conformation spatial arrangements of a molecule's atoms are vital for determining chemical behavior and reactivity (Axelrod & Gomez-Bombarelli, 2022). Traditional methods, such as RDKit (Riniker & Landrum, 2015), utilize experimental torsion knowledge and distance geometry to manipulate torsion angles and explore conformational space (Kang et al., 1996; Havel, 1998). These approaches have significantly contributed to the field by enhancing our ability to predict and analyze molecular structures (Hawkins, 2017). However, they often face prohibitive computational costs

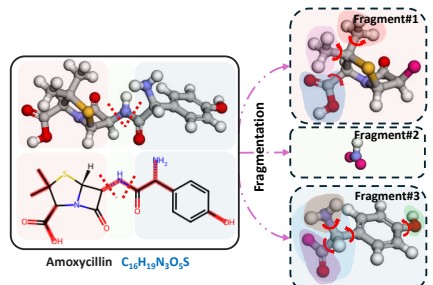

Figure 1: Fragmentation example on *Amoxicillin*.

and limited generalizability with large and complex molecules, as exploring all possible conformations is expensive and these methods may not generalize well across diverse molecular systems (Rappé et al., 1992; Halgren, 1996; Zhou et al., 2023).

On the other hand, data-driven generation methods have recently gained prominence due to their remarkable ability to capture the overall structural distribution of molecules (Gómez-Bombarelli et al., 2018; Fu et al., 2020; 2021; Hoffman et al., 2022), wherein diffusion-based generative models stand out with exemplary performance (Guo et al., 2024; Xu et al., 2022; Wang et al., 2023). The high-level idea of diffusion-based molecular generation methods undergoes a transition from stable equilibrium conformations to a state of increased disorder through a series of controlled diffusion steps, and then learns a model to reverse the diffusion process (Song & Ermon, 2019; Ho et al., 2020; Xu et al., 2022). These methods leverage vast datasets to learn and predict molecu-

lar structures (Ryan et al., 2018; Ross et al., 2022; Siebenmorgen et al., 2024). Diffusion models that operate 3D Euclidean space can struggle with computational efficiency and scalability when dealing with large, complex molecules (Xu et al., 2022). To overcome this problem, torsional diffusion (Jing et al., 2022) focuses specifically on the torsion angles of molecules, efficiently exploring conformational space by leveraging the periodic nature of these angles. This approach reduces the dimensionality of the problem and allows for more targeted exploration of low-energy conformers, making it well-suited for molecular conformation generation task. Despite the advancements brought about by torsional diffusion, challenges remain, particularly in terms of *data efficiency and generalization* (Wang et al., 2022; Tiwary et al., 2024). One of the primary limitations of current diffusion-based methods is their heavy reliance on large, high-quality datasets to learn molecular structures (Heid et al., 2023; Rotskoff, 2024; Tiwary et al., 2024). In practice, obtaining such datasets can be costly and time-consuming, especially for complex molecules or novel chemical spaces (Huang & Von Lilienfeld, 2021). The limited data efficiency of these methods further constrains the models' generalization ability and expressive power (Kirchmeyer et al., 2022; Tiwary et al., 2024). This reliance on extensive training data can lead to suboptimal performance when applied to molecules that deviate significantly from those in the training set (Rotskoff, 2024).

To address these limitations, we propose a fragment-based data augmentation strategy which leverages the modularity of molecules for molecular conformer generation within diffusion models. By decomposing complex molecules into smaller fragments, we can generate diverse conformations for each substructure (Gordon et al., 2012). For instance, *Amoxicillin* in Fig. 1, a complex antibiotic, can be fragmented into key components such as the *β-lactam ring* and *thiazolidine ring*, an *amino group*, and a *hydroxyl group* along with a *benzene ring*. These fragments augment the dataset with a wide range of fragment-level configurations, increasing data diversity while capturing local structural variations under chemical priors. Such properties are common in chemical spaces, where functional groups or substructures exhibit consistent behavior across different molecules (Liu et al., 2017; Jinsong et al., 2024). By incorporating fragment-level semantics, we exploit molecular regularity to improve the model's generalization while ensuring that generated conformations remain chemically valid by preserving essential structural and torsional properties (Horton et al., 2022).

In summary, our contributions are as follows: We propose a fragment-augmented diffusion framework FADiff for molecular conformation generation, leveraging the inherent modularity of molecules with fragment-based data augmentation. This approach increases data diversity and captures local structural variations under chemical priors, thereby enhancing the data efficiency and generalization capabilities of diffusion-based generative models, particularly for large and complex molecular systems. We provide an in-depth theoretical analysis showing how our fragment-based strategy improves model performance, shedding light on the underlying mechanisms that contribute to its effectiveness. Additionally, we conduct extensive experiments that demonstrate the superior performance of our method over existing approaches. Our strategy maintains chemical validity and integrity by preserving essential structural and torsional properties, which improves the exploration of conformational space and benefits molecular conformation generation tasks.

## 2 RELATED WORK

**Diffusion-based Molecular Generation**   Diffusion models have gained significant attraction as powerful tools for drug discovery applications (Xu et al., 2022; Guo et al., 2024; Hua et al., 2024). These methods typically start by transitioning from stable equilibrium conformations to a state of heightened disorder via a sequence of regulated diffusion steps (Yang et al., 2023; Guo et al., 2024; Cao et al., 2024), with a model trained to reverse this process (Xu et al., 2022). Recent methods for molecular conformation generation model directly in 3D Euclidean space, employing equivariant graph neural networks within diffusion models to process atomic coordinates and features (Xu et al., 2022; Hoogeboom et al., 2022). These approaches inject Gaussian noise into all spatial coordinates, requiring numerous denoising steps (Shi et al., 2021; Luo et al., 2021; Xu et al., 2022). Recognizing that molecular flexibility arises primarily from torsional degrees of freedom (Axelrod & Gomez-Bombarelli, 2022; Jing et al., 2022) proposes Torsional Diffusion, wherein the diffusion process acts only on torsion angles while keeping other degrees of freedom fixed. Focusing on torsion angles reduces the dimensionality of the sample space, leading to more efficient and effective conformer generation. It leverages torsion angles to model the potential energy surface, a fundamental

component in computational chemistry (Kang et al., 1996), by combining diffusion processes with detailed torsional angle modeling to enhance both accuracy and efficiency in conformer generation.

**Molecular Fragment Decomposition**   Molecular fragment decomposition is a critical concept in computational chemistry, enabling the simplification of complex molecular structures into smaller and more manageable units (Hann et al., 2001; Sliwoski et al., 2014; Sadybekov & Katritch, 2023). This approach facilitates the study of molecular properties and interactions by focusing on individual fragments that retain key chemical characteristics of the parent molecule (Bemis & Murcko, 1996; Jinsong et al., 2024). From a force field perspective, by preserving the local chemical environment around targeted torsions, fragmentation allows for accurate modeling of torsional potentials, ensuring that torsional characteristics can be effectively transferred back to the parent molecule (Horton et al., 2022; D'Amore et al., 2022). Fragmentation rules, such as BRICS (Breaking of Retrosynthetically Interesting Chemical Substructures), allow for the systematic breakdown of molecules based on chemically meaningful bonds, preserving functional groups that are essential for chemical activity (Lewell et al., 1998; Degen et al., 2008). This decomposition not only aids in understanding the intrinsic properties of molecular subunits but also enhances computational efficiency by reducing the complexity of conformational space (Liu et al., 2017). By analyzing these fragments, researchers can predict reactivity, optimize drug design, and explore novel chemical spaces with higher precision (Gordon et al., 2012). The integration of molecular fragment decomposition with advanced modeling techniques, such as torsional diffusion, offers a powerful framework for generating accurate and diverse molecular conformers, ultimately advancing the fields of drug discovery and materials science (Jinsong et al., 2024).

## 3 METHODOLOGY

### 3.1 PRELIMINARIES

**Notations and Problem Formulations**   Each molecule with $n$ atoms is represented as an undirected graph $\mathcal{G} = (\mathcal{V}, \mathcal{E})$, where $\mathcal{V} = \{v_i\}_{i=1}^n$ represents the atoms and $\mathcal{E} = \{e_{i,j} \mid (i,j) \subseteq \mathcal{V} \times \mathcal{V}\}$ represents the bonds between atoms. Each node $v_i$ describes atomic attributes, such as element type, and each edge $e_{i,j}$ describes the bond between $v_i$ and $v_j$, labeled with its chemical type. The goal of molecular conformation generation is to generate stable conformations for a given molecular graph $\mathcal{G}$. While conformations $C$ can be described by atomic positions, it is often more efficient to use internal coordinates like bond lengths, bond angles, and torsion angles. Torsion angles are particularly important as they capture the rotations around rotatable bonds, which define the molecule's flexibility. Each rotatable bond introduces a degree of freedom, corresponding to a torsion angle. By focusing on torsion angles, we reduce the problem's dimensionality while preserving the molecule's conformational flexibility. Thus, conformations $C$ are represented as a set of torsion angles $\boldsymbol{\tau} = \{\tau^i\}_{i=1}^m$, where $m$ is the number of rotatable bonds. These torsion angles define the 3D structure in torsional space $\mathbb{T}^m$. For each molecular graph $\mathcal{G}$, its conformations $C$ are treated as i.i.d. samples from an underlying Boltzmann distribution.

**Torsion Computation Basics**   Directly learning a score model over intrinsic torsion coordinates presents several challenges. First, the dimensionality of the torsional space depends on the molecular graph $\mathcal{G}$, and the mapping from torsional space to conformers is influenced by both $\mathcal{G}$ and local structures $L$. Additionally, there is no canonical way to define torsion angles, as they depend on arbitrary choices of reference neighbors. To address these issues, (Jing et al., 2022) represent conformers as 3D point clouds in extrinsic coordinates, which are invariant to global roto-translation. This allows us to construct a score model $\mathbf{s}_\theta(C, t)$ that operates over 3D conformers while predicting updates in the torsional space. By applying torsion updates directly to the 3D coordinates, we avoid the need to define reference neighbors, ensuring invariance to such choices (Quack, 2002). Furthermore, the model must respect parity equivariance, meaning the learned score function must change sign under parity inversion, ensuring that the model outputs pseudoscalars that are invariant under $SE(3)$ transformations but change sign when the input point cloud is inverted (Jing et al., 2022). More details on torsion angle invariance and parity equivariance can be found in the Appendix A.1.

**Torsional Diffusion Basics**   Diffusion-based models have emerged as powerful tools for molecular conformation generation, particularly in drug discovery (Xu et al., 2022; Guo et al., 2024). These

models use stochastic differential equations (SDEs) to transition molecular structures from stable conformations to disorder, with the reverse process generating samples from the data distribution. Specifically, torsional diffusion focuses on modeling the torsion angles of a molecule, which define a hypertorus $\mathbb{T}^m$ with each angle in $[0, 2\pi)$ (De Bortoli, 2022; Jing et al., 2022). This approach efficiently explores conformational space by leveraging the intrinsic properties of torsion angles and adapting diffusion models to compact Riemannian manifolds. The forward and reverse process can be described as follows:

**Forward Process**  In the torsional diffusion model, the forward process gradually adds noise to the torsion angles $\tau$ of a molecular conformer over time $t \in [0, T]$. The perturbation of torsion angles is modeled as a wrapped normal distribution, ensuring that the periodic nature of torsion angles is respected. Specifically, the distribution of the perturbed torsion angles $\tau'$ given the previous angles $\tau$ is:

$$p_{t|0}(\tau' \mid \tau) \propto \sum_{d \in \mathbb{Z}^m} \exp\left(-\frac{\|\tau' - \tau + 2\pi d\|^2}{2\sigma^2(t)}\right),$$

where $\sigma(t)$ controls the scale of the noise at each time step. As time progresses, the noise injected into the system increases, and by the final time step $T$, the distribution $p_T(\tau)$ approaches a Gaussian, representing a highly disordered state. The noise scale $\sigma(t)$ is defined as: $\sigma(t) = \sigma_{\min} e^{t \log \frac{\sigma_{\max}}{\sigma_{\min}}}$, where $\sigma_{\min} = 0.01\pi$ and $\sigma_{\max} = \pi$. This time-dependent variance function ensures that the amount of noise increases smoothly over time, allowing the model to explore the torsional space of the molecule by gradually perturbing the torsion angles (De Bortoli, 2022; Jing et al., 2022).

**Reverse Process**  The reverse process generates molecular conformations by reversing the forward diffusion process, starting from a noisy state $\tau^T \sim p_T(\tau)$ and iteratively refining it to recover a stable conformation $\tau^0$. At each step, the reverse process denoises the torsion angles, progressively moving them from a disordered state back to a stable configuration. The reverse transitions are guided by the score function $\nabla_\tau \log p_t(\tau)$, which directs the system towards the data distribution. The score function $\nabla_\tau \log p_t(\tau)$ is approximated by a neural network $\mathbf{s}(\tau, t)$, which is trained to match the true score function. The network $\mathbf{s}_\theta(C, t)$ represents the gradient of the log-probability of the perturbed torsion angles at time $t$, and it is learned during training using denoising score matching (DSM) (Ho et al., 2020). The loss function for DSM is defined as:

$$J_{\text{DSM}}(\theta) = \mathbb{E}_t \left[\lambda(t) \mathbb{E}_{\tau^0 \sim p_0, \tau \sim p_{t|0}(\cdot|\tau^0)} \left[\|\mathbf{s}_\theta(C, t) - \nabla_\tau \log p_{t|0}(\tau \mid \tau^0, \mathcal{G})\|^2\right]\right],$$

where $\mathbf{s}_\theta(C, t)$ is the neural network that approximates the score function, and $\lambda(t)$ are precomputed weight factors that balance the contribution of different time steps. This reverse process iteratively refines the torsion angles, leveraging the learned score function $\mathbf{s}_\theta(C, t)$ to recover stable molecular conformations from noisy initial states.

### 3.2 TORSIONAL SCORE DIFFUSION BACKBONE NETWORK

The proposed backbone network is built on the tensor product convolutional layer (Jing et al., 2022), which integrates node features, edge features, and geometric information (e.g., spherical harmonics) through tensor product operations (Thomas et al., 2018; Geiger & Smidt, 2022). This design ensures equivariance w.r.t. both rotations and translations, crucial for molecular generation tasks. The network predicts a pseudoscalar $\Delta\tau$ for each rotatable bond, used in the torsional score diffusion process. To handle rotational symmetries, we represent the directionality of edges using spherical harmonics $\mathbf{Y}_{ij}$ (Jing et al., 2022). These harmonics are derived from $\mathbf{r}_{ij}$, the relative position vector between atoms $i$ and $j$ that defines the edge direction (Kondor et al., 2018). By encoding the edge direction with $\mathbf{Y}_{ij}$, it ensures that our model appropriately accounts for rotational symmetries. The network updates node and edge features through tensor product operations that combine node features $\mathbf{h}_i^l$ and edge features $e_{ij}^l$ at layer $l$, along with spherical harmonics $\mathbf{Y}_{ij}$. Here, $l$ denotes the layer index in the network. The update process can be represented as:

$$\mathbf{h}_i^{l+1} = \sum_{j \in \mathcal{N}(i)} \mathbf{W}_l \left(\mathbf{h}_j^l \otimes \mathbf{Y}_{ij} \otimes \mathbf{W}_e e_{ij}^l\right),$$

where $\mathcal{N}(i)$ denotes the set of neighbors of node $i$, $\mathbf{W}_l$ is the learnable weight matrix for layer $l$, and $\mathbf{W}_e$ is the weight matrix that maps edge features. $\otimes$ denotes the tensor product operation.

Figure 2: An overview of the Fragment-Augmentated Diffusion (FADiff) pipeline. Molecules from the training set will be fragmented based on randomly selected fragmentation edges. The resulting fragments will be further used as augmented data in the training phase.

This ensures that updated edge features remain equivariant to rotations. After the final convolutional layer, edge features are processed to generate the final edge representation, used to predict the pseudoscalar for each rotatable bond:

$$\Delta\hat{\tau}^{i,j} = \mathbf{W}_O e_{ij}^L, \quad e_{ij}^L = \mathbf{W}_L \left( \mathbf{h}_i^L + \mathbf{h}_j^L \right),$$

where $\mathbf{W}_L$ is the learnable weight matrix corresponding to the last layer $L$, and $\mathbf{W}_O$ is the output projection weight matrix. This process ensures effective torsional score prediction for each rotatable bond, while maintaining equivariance to both rotations and translations. Further details on torsional score diffusion backbone network are provided in the Appendix A.1.

## 3.3 FRAGMENT-AUGMENTED DIFFUSION

In this work, we propose a fragment-based augmentation approach for diffusion generation model, where molecules are decomposed into smaller meaningful fragments using specific rules. This decomposition is guided by identifying key rotatable bonds or functional groups, ensuring that each fragment retains essential chemical and structural information (Jinsong et al., 2024). By focusing on these fragments, we aim to enhance the torsion diffusion process for molecular conformation generation. This method allows the model to handle smaller, more manageable substructures, which can be optimized independently, while maintaining global molecular consistency through interactions between fragments. Once a molecule is decomposed into multiple fragments, each fragment is treated as an independent subgraph consisting of its own nodes (atoms) and edges (bonds), and the conformation generation task for each fragment is performed independently. Molecules from the training set are fragmented based on randomly selected fragmentation edges. The resulting smaller fragments are used as augmented data in the training phase. The loss for each fragment is computed separately, and these losses are summed to form the total loss function. Specifically, if a molecule is decomposed into $B + 1$ fragments (via $B$ decomposition edges), the total loss can be expressed as:

$$\mathcal{L}_{\text{total}} = \frac{1}{B+1} \sum_{b=1}^{B+1} \mathbb{E}_{(u,v)\in\mathcal{E}_b} \left[ \|\mathbf{s}_\theta(C, t)^{u,v} - \nabla_\tau \log p_{t|0}(\tau \mid \tau^0, \mathcal{G}_b)\Big|_{\tau=\boldsymbol{\tau}_b^{u,v}}\|^2 \right],$$

$\mathcal{E}_b$ represents the set of edges in the $b$-th fragment, and $\tau^{u,v}$ denotes the torsion angle associated with the bond between nodes $u$ and $v$. $\mathbf{s}_\theta(C, t)$ is the predicted score for the torsion angles at time $t$, and $\nabla_\tau \log p_{t|0}(\tau \mid \tau^0, \mathcal{G}_b)$ is the actual gradient of the torsion angles, conditioned on the initial state $\tau^0$ and the local graph structure $\mathcal{G}_b$ of the fragment. Figure 2 illustrates an overview of the fragment augmentation diffusion pipeline.

## 3.4 CONNECTING FRAGMENT-BASED MOLECULAR MODELING TO DATA AUGMENTATION

In fragment-based molecular modeling, our goal is to enhance the diversity of molecular representations while retaining key torsional properties. By employing fragmentation methods to decompose the complete molecular torsion space $\boldsymbol{\tau}$ into smaller torsional subspaces $\{\boldsymbol{\tau}_b\}_{b=1}^{B+1}$, we generate multiple views of a molecule's torsional characteristics. Each torsional subspace $\boldsymbol{\tau}_b = \{\tau_b^{u,v}\}_{(u,v)\in\mathcal{E}_b}$ retains important torsional and geometric information from the complete torsion space $\boldsymbol{\tau}$. In practical applications, due to data limitations, the true fragment torsional angles $\boldsymbol{\tau}_b$ are often unavailable.

To address this issue, our proposed data augmentation strategy uses the torsional angles $\hat{\tau}_b$ from the complete molecular structure $\boldsymbol{\tau} = \{\hat{\tau}_b\}_{b=1}^{B+1}$ as approximations. This approximation is based on the assumption that, due to the preserved local chemical environment, the torsional properties of the fragments are very similar to those of the complete molecule, that is, $\boldsymbol{\tau}_b \approx \hat{\tau}_b$. However, the assumption does not always hold because fragmenting the molecule may alter the electronic environment and interactions, leading to differences in torsional properties between the fragments and the complete molecule (Stern et al., 2022). Such differences may introduce errors in the modeling process. Additionally, using the complete molecule's torsional angles as approximations for the fragments ignores potential conformational changes that may occur in the fragments due to the absence of steric hindrance or electronic interactions present in the complete molecule (Horton et al., 2022). Thus, the core idea is that fragmenting molecules while preserving the local chemical environment allows the torsional properties of the fragments to remain consistent with those of the complete molecule, thus ensuring torsional consistency between $\hat{\tau}_b$ and $\boldsymbol{\tau}_b$ (Stern et al., 2022; Horton et al., 2022).

We can further consider this problem from the perspective of mutual information. By employing appropriate fragmentation methods, we aim to maximize the mutual information (MI) between $\hat{\boldsymbol{\tau}}$ and $\boldsymbol{\tau}$, ensuring that the fragment torsional subspaces retain sufficient global torsional information to accurately reflect molecular properties such as conformational flexibility and stability. The mutual information is expressed as:

$$I(\hat{\boldsymbol{\tau}}_b; \boldsymbol{\tau}_b) = H(\boldsymbol{\tau}_b) - H(\boldsymbol{\tau}_b \mid \hat{\boldsymbol{\tau}}_b),$$

where $H(\boldsymbol{\tau}_b)$ is the entropy of the fragment torsional subspace, and $H(\boldsymbol{\tau}_b \mid \hat{\boldsymbol{\tau}}_b)$ is the conditional entropy of the fragment torsional subspace given the complete torsion space. By minimizing the conditional entropy $H(\boldsymbol{\tau}_b \mid \hat{\boldsymbol{\tau}}_b)$, we effectively maximize the mutual information, ensuring that the torsional angles of the complete molecule can accurately predict those of the fragments, thus enhancing model performance. To formalize the relationship between fragmentation strategy and mutual information, we introduce the following lemma:

**Lemma 1** *Let $\zeta^* = \arg\max_{\zeta \in \mathcal{F}} I_\zeta(\hat{\boldsymbol{\tau}}; \boldsymbol{\tau})$, where $\mathcal{F}$ is the space of all possible fragmentation strategies, and $I_\zeta(\hat{\boldsymbol{\tau}}; \boldsymbol{\tau})$ denotes the mutual information between $\hat{\boldsymbol{\tau}}$ and $\boldsymbol{\tau}$ under fragmentation strategy $\zeta$. Then, the fragmentation strategy $\zeta^*$ that maximizes $I_\zeta(\hat{\boldsymbol{\tau}}; \boldsymbol{\tau})$ is the optimal strategy that enhances the torsional information retention:*

$$I_{\zeta^*}(\hat{\boldsymbol{\tau}}; \boldsymbol{\tau}) = \max_\zeta I_\zeta(\hat{\boldsymbol{\tau}}; \boldsymbol{\tau}).$$

This lemma demonstrates that selecting the optimal fragmentation strategy—by maximizing the mutual information $I_\zeta(\hat{\boldsymbol{\tau}}; \boldsymbol{\tau})$ ensures that fragments retain relevant torsional and geometric information from $\hat{\boldsymbol{\tau}}$. Thus, by focusing on chemically meaningful torsion subspaces, fragment-based torsion modeling ensures that local fragment optimizations contribute to a globally consistent molecular conformation. More discussion of this lemma is provided in the Appendix A.5.

Therefore, we need to carefully choose fragmentation methods to ensure the preservation of key chemical properties and be aware of the biases this approximation may introduce. By recognizing these limitations and thoughtfully considering fragmentation strategies, we can mitigate potential errors and effectively leverage the advantages of data augmentation in fragment-based molecular modeling, thereby improving the robustness and accuracy of the model.

**Error Analysis** During molecular fragmentation, chemical or graph-based rules decompose the molecule into smaller fragments. The choice of decomposition edges introduces errors between the torsion angles $\hat{\tau}_b^{u,v}$ from the full molecular graph and the true torsion angles $\tau_b^{u,v}$ of the fragments due to potential loss of structural or torsional information. Assuming that the error $\epsilon$ is a random variable with probability density function $p(\epsilon)$, we can express the conditional entropy $H(\tau_b^{u,v} \mid \hat{\tau}_b^{u,v})$ in terms of the differential entropy of the error $\epsilon$:

$$H(\tau_b^{u,v} \mid \hat{\tau}_b^{u,v}) = h(\epsilon),$$

where $h(\epsilon)$ denotes the differential entropy of $\epsilon$. This is because, given $\tau_b^{u,v} = \hat{\tau}_b^{u,v} + \epsilon$, the uncertainty in $\tau_b^{u,v}$ when $\hat{\tau}_b^{u,v}$ is known is entirely due to the uncertainty in $\epsilon$. If we model the error $\epsilon$ as a zero-mean Gaussian random variable with variance $\sigma^2$ (i.e., $\epsilon \sim \mathcal{N}(0, \sigma^2)$), its differential entropy is: $h(\epsilon) = \frac{1}{2}\ln(2\pi e\sigma^2)$, and we have

$$I(\hat{\tau}_b^{u,v}; \tau_b^{u,v}) = H(\tau_b^{u,v}) - h(\epsilon) = H(\tau_b^{u,v}) - \frac{1}{2}\ln(2\pi e\sigma^2).$$

This equation reveals that as the error variance $\sigma^2$ increases, the mutual information $I(\hat{\tau}_b^{u,v}; \tau_b^{u,v})$ decreases, indicating that the association between $\hat{\tau}_b^{u,v}$ and $\tau_b^{u,v}$ becomes weaker. Different fragmentation methods influence the error variance $\sigma^2$, and consequently, the mutual information between $\hat{\tau}_b^{u,v}$ and $\tau_b^{u,v}$. For example, fragmentation methods that preserve key chemical factors such as conjugation, resonance, steric hindrance, and hydrogen-bonding interactions can reduce $\sigma^2$, thereby increasing the mutual information and enhancing model accuracy. Building upon the previous error analysis and the lemma presented, we can further explore how the choice of fragmentation strategy affects the error bound and model performance by providing a lower bound on the error variance $\sigma^2$ achievable by any fragmentation strategy as: $\sigma^2 \geq \sigma_{\zeta*}^2$. We provide a detailed analysis of how different fragmentation methods affect $\sigma^2$ by considering these factors in Appendix B.4.

## 3.5 TRAINING OBJECTIVE

**Variational Lower Bound (ELBO) Optimization**  Using the probability flow ODE, we compute the likelihood of any sample $\tau$ as $\log p_0(\tau^0) = \log p_T(\tau^T) - \frac{1}{2}\int_0^T \frac{d}{dt}\sigma^2(t)\nabla_\tau \cdot \mathbf{s}_\theta(C,t)\,dt$. (Song & Ermon, 2020; De Bortoli, 2022). Since directly optimizing the exact log-likelihood is intractable, we instead maximize the usual variational lower bound (ELBO), which provides a tractable approximation to the log-likelihood as:

$$\mathbb{E}[\log p_\theta(\tau^0|\mathcal{G})] = \mathbb{E}\left[\log \frac{p_\theta(\tau^{0:T}|\mathcal{G})}{q(\tau^{1:T}|\tau^0)}\right] \geq -\mathbb{E}_q\left[\sum_{t=1}^T D_{\mathrm{KL}}(q(\tau^{t-1}|\tau^t,\tau^0)\|p_\theta(\tau^{t-1}|\tau^t,\mathcal{G}))\right],$$

where $q(\tau^{t-1}|\tau^t,\tau^0)$ is analytically tractable as $\mathcal{N}\left(\sqrt{\frac{\alpha_t-\beta_t}{1-\alpha_t}}\tau^0 + \frac{\sqrt{\alpha_t(1-\alpha_t)}}{1-\alpha_t}\tau^t, \frac{1-\alpha_t}{1-\alpha_t}\beta_t\right)$. $\alpha_t = 1 - \beta_t$ and $\bar{\alpha}_t = \prod_{s=1}^t \alpha_s$ are derived from the special property of the forward process, where $q(\tau^t|\tau^0)$ of arbitrary timestep $t$ can be calculated in closed form $q(\tau^t|\tau^0) = \mathcal{N}(\tau^t; \sqrt{\bar{\alpha}_t}\tau^0, (1-\bar{\alpha}_t)I)$. This indicates with sufficiently large $T$, the whole forward process will convert $\tau^0$ to a whitened isotropic Gaussian, and thus it is natural to set $p(\tau^T)$ as a standard Gaussian distribution. The complete derivation of the ELBO is provided at Appendix A.4.

**Energy-based Training**  By maximizing likelihoods with ELBO, we can train models to match the Boltzmann distribution over torsion angles. The energy-based training consists of resampling and score matching stages. In resampling, the model acts as an importance sampler using the torsional Boltzmann density $\tilde{p}(\tau \mid \mathcal{G})$. In score matching, importance weights approximate the denoising score-matching loss. We sample torsion angles $\tau^1, \ldots, \tau^K \sim q(\tau \mid \mathcal{G})$ from the torsional diffusion model. and we compute the importance weight $\tilde{w}_k = \tilde{p}_\theta(\tau^k \mid \mathcal{G})/q(\tau^k \mid \mathcal{G})$ for each sample $\tau^k$. These weights are used to approximate the denoising score matching loss $J_{DSM}$ for $p_0 \propto \tilde{p}$. The objective is to minimize the weighted loss:

$$J_{DSM}(\theta) = \mathbb{E}_t\left[\lambda(t)\mathbb{E}_{\tau^0 \sim p_0, \tau^t \sim p_{t|0}(\cdot|\tau^0)}\left[\tilde{w}(\tau^t)\|\mathbf{s}_\theta(C,t) - \nabla_\tau \log p_{t|0}(\tau^t \mid \tau^0, \mathcal{G})\|^2\right]\right], \quad (1)$$

where the importance weights $\tilde{w}_k$ adjust the contribution of each sample to the loss, ensuring that the model learns to generate samples that are consistent with the Boltzmann distribution.

## 4 EXPERIMENTS

### 4.1 EXPERIMENTAL SETUP

**Dataset**  We follow the datasets used in (Jing et al., 2022), which include 3 subsets from the GEOM dataset (Axelrod & Gomez-Bombarelli, 2022). The GEOM dataset provides high-quality conformation ensembles generated using metadynamics in CREST (Pracht et al., 2020). Specifically, we utilize GEOM-QM9, GEOM-DRUGS, and GEOM-XL. GEOM-QM9 is a dataset featuring significantly smaller molecules with an average of 11 atoms. GEOM-DRUGS represents the most pharmaceutically relevant subset, comprising molecules with an average of 44 atoms. GEOM-XL is created by selecting all species with more than 100 atoms from GEOM-MoleculeNet (Wu et al., 2018), allowing us to evaluate models' generation quality on large molecules. For a detailed statistics for all three datasets are in Appendix D.2.

**Evaluation** We evaluate the quality of the generated conformation ensembles using the train/val/test splits setup from Jing et al. (2022) and apply RMSD-based metrics to assess both diversity and quality. The key metrics include Average Minimum RMSD (AMR) and Coverage (COV), which are reported for both Recall (AMR-R, COV-R) and Precision (AMR-P, COV-P). COV-R and AMR-R, measure how well the generated ensemble covers the ground-truth ensemble, while COV-P and AMR-P, assess the accuracy of the generated conformers. The calculation of COV-R and AMR-R can be defined as:

$$\text{COV-R} := \frac{1}{L} \left| \{ l \in [1..L] : \exists k \in [1..K], \text{RMSD}(C_k, C_l^*) < \delta \} \right|$$

$$\text{AMR-R} := \frac{1}{L} \sum_{l \in [1..L]} \min_{k \in [1..K]} \text{RMSD}(C_k, C_l^*)$$

For precision, COV-P and AMR-P are calculated by swapping the roles of generated and reference sets. These metrics emphasize the quality of generated conformations, with $\delta$ set to 0.5Å for GEOM-QM9 and 0.75Å for GEOM-DRUGS datasets evaluation. Higher COV or lower AMR scores suggest more realistic conformations, balancing both diversity and quality.

**Baselines** We compare FADiff with methods from both traditional computational methods and established state-of-the-art deep learning baselines. Among traditional computational methods, we employ RDKit ETKDG (Havel, 1998; Riniker & Landrum, 2015), the most established open-source package, and OMEGA (Hawkins, 2017), a commercial software in continuous development. Among deep learning methods, we evaluate CGCF (Xu et al., 2021b), ConfVAE (Xu et al., 2021a), ConfGF (Shi et al., 2021), GeoMol (Ganea et al., 2021), Geodiff (Xu et al., 2022), and Torsional Diffusion (For simplicity, we name it as TorDiff in the subsequent section) (Jing et al., 2022).

**Fragmentation Augmentation** For a given molecule, we identify all fragmentation-edges and randomly select $B = \min(b, \kappa)$ edges, where $b$ is the total number of fragmentation-edges and $\kappa$ limits the maximum number of selected edges to avoid excessive small fragments. From the resulting $B + 1$ fragments, those with rotatable bonds are used to augment the training set. Our experiments use $\kappa = 5$. During fragmentation, only fragments larger than $z$ atoms are selected for augmentation. This ensures that the resulting fragments retain sufficient structural complexity and chemical information to contribute meaningfully to the training process. To explore the impact of reaction-related bonds on model performance, we also test models generated after removing these bonds, focusing on BRICS and RECAP rules (Lewell et al., 1998; Degen et al., 2008). Detailed introductions of these two chenmical rules are provided in the Appendix B.1, and Additional results and discussions on how the choice of the minimum fragment size parameter $z$ affects fragmentation statistics are provided in Appendix D.1.

**Computational-Aided Data Augmentation** The conformer matching method mitigates the distributional shift between training and test time by aligning ground truth conformers with synthetic ones generated by RDKit, ensuring consistency between the two distributions. In Jing et al. (2022), training on these synthetic conformers has shown significant better performance than using ground truth alone. Therefore, we use conformer matching as a additional data augmentation technique, generating synthetic proxy conformers alongside the original ones. Details and ablation studies are provided in Appendix C.

## 4.2 CONFORMATION GENERATION

As shown in Table 1, FADiff outperforms other models in both coverage and RMSD metrics. Specifically, FADiff achieves the highest mean COV-R (70.07%) and COV-P (52.87%), indicating its ability to generate a wide range of conformers that cover the conformational space effectively. Additionally, FADiff exhibits the lowest mean AMR-R (0.609 Å) and AMR-P (0.588 Å), reflecting its precision in generating conformers that closely match the reference structures. Compared to other methods like TorDiff and GeoDiff, FADiff consistently delivers superior performance across all metrics, particularly excelling in both coverage and accuracy. This highlights the effectiveness of the fragment-augmentation strategy in exploring the conformational space.

Table 1: Quality of generated conformation ensembles for the GEOM-DRUGS test set in terms of Coverage (%) and Average Minimum RMSD (Å) with $\delta = 0.75$ Å.

| Models | COV-R (%) ↑ | | AMR-R (Å) ↓ | | COV-P (%) ↑ | | AMR-P (Å) ↓ | |
| --- | --- | --- | --- | --- | --- | --- | --- | --- |
| | Mean | Median | Mean | Median | Mean | Median | Mean | Median |
| Metrization | 5.71 | 0.00 | 1.388 | 1.329 | 4.932 | 0.000 | 1.541 | 1.339 |
| CGCF | 19.13 | 12.53 | 1.248 | 1.224 | 1.682 | 0.000 | 1.857 | 1.806 |
| ConfVAE | 14.01 | 14.83 | 1.238 | 1.141 | 2.963 | 0.000 | 1.828 | 1.815 |
| ConfGF | 15.15 | 11.93 | 1.162 | 1.159 | 2.425 | 0.000 | 1.721 | 1.686 |
| GeoMol | 34.19 | 26.45 | 1.087 | 1.058 | 20.66 | 15.07 | 1.184 | 1.110 |
| OMEGA | 53.40 | 54.60 | 0.841 | 0.762 | 40.50 | 33.30 | 0.946 | 0.854 |
| ETKDG | 38.40 | 28.60 | 1.058 | 1.002 | 40.90 | 30.80 | 0.995 | 0.895 |
| GeoDiff | 45.61 | 49.32 | 0.862 | 0.852 | 21.47 | 14.55 | 1.171 | 1.123 |
| TorDiff | 67.49 | 75.81 | 0.634 | 0.618 | 49.53 | 47.16 | 0.827 | 0.778 |
| FADiff | **70.07** | **78.35** | **0.609** | **0.588** | **52.87** | **54.17** | **0.800** | **0.749** |

Table 3: Quality of generated conformation ensembles for the GEOM-DRUGS test set with $\delta = 0.75$ Å on varying available training samples **n**.

| Models | | FADiff | | | | TorDiff | | | |
| --- | --- | --- | --- | --- | --- | --- | --- | --- | --- |
| Metric | | COV-R | AMR-R | COV-P | AMR-P | COV-R | AMR-R | COV-P | AMR-P |
| | 1000 | **49.39** | **0.7928** | **33.84** | **1.0455** | 34.60 | 0.8933 | 20.84 | 1.1897 |
| **n** | 5000 | **51.17** | **0.7519** | **34.51** | **1.0389** | 44.61 | 0.8209 | 25.77 | 1.1104 |
| | 10000 | **62.82** | **0.6736** | **43.10** | **0.9081** | 52.76 | 0.7507 | 33.88 | 1.0458 |

**Performance on Large Molecule Generation**
We further evaluate our method on the GEOM-XL dataset, which contains molecules with an average number of atoms approximately three times larger than those in the GEOM-Drugs dataset used for training. The results are presented in Table 2. Our model, FADiff, significantly outperforms TorDiff and other baseline

Table 2: Performance on the GEOM-XL dataset.

| Model | AMR-R ↓ | | AMR-P ↓ | |
| --- | --- | --- | --- | --- |
| | Mean | Med | Mean | Med |
| RDKit | 2.92 | 2.62 | 3.35 | 3.15 |
| GeoMol | 2.47 | 2.39 | 3.30 | 3.15 |
| TorDiff | 2.05 | 1.86 | 2.94 | 2.78 |
| FADiff | **1.80** | **1.61** | **2.60** | **2.44** |

models in generating conformations for large molecules. Specifically, FADiff achieves the lowest mean AMR-R of 1.80 Å and median AMR-R of 1.61 Å, indicating superior recall performance. Similarly, it attains the lowest mean AMR-P of 2.60 Å and median AMR-P of 2.44 Å, demonstrating better precision in generating conformations close to the reference structures. These improvements highlight the effectiveness of our fragment-based data augmentation strategy in enhancing the generalization capabilities of diffusion models to larger and more complex molecular systems.

**Model Performance Across Different Training Sample Sizes** Table 3 illustrates that FADiff consistently outperforms TorDiff across all metrics and training sample sizes. For 1000 samples, FADiff achieves a COV-R of 49.39%, which is 42% higher than TorDiff, and reduces AMR-R to 0.7928 Å compared to TorDiff's 0.8933 Å. When the training size increases to 5000 samples, FADiff maintains its advantage with a COV-R of 51.17%, outperforming TorDiff's 44.61%, and lowering AMR-R to 0.7519 Å from 0.8209 Å. At the largest sample size of 10000 samples, FADiff achieves a COV-R of 62.82%, which is 19% higher than TorDiff, and further decreases AMR-R to 0.6736 Å *v.s.* TorDiff's 0.7507 Å. Additionally, FADiff also consistently delivers superior performance in COV-P and AMR-P metrics across all sample sizes, demonstrating its enhanced capability to generate conformation ensembles that are not only diverse but also accurate with limited data. In all, these results highlight FADiff's robustness and scalability, particularly in data-scarce environments, while also demonstrating its capacity to improve with larger datasets.

**Impact of Chemical Fragmentation** Table 4 shows the effect of removing BRICS and RECAP reaction edges on conformer generation. The full FADiff model, which includes both, achieves the best performance with a mean COV-R of 51.17% and COV-P of 50.10%. Removing BRICS (w/o BRICS) has a larger impact on precision, reducing COV-P to 33.93% and increasing AMR-P to 1.0461 Å, indicating BRICS edges are crucial for precision. In contrast, removing RECAP (w/o RECAP) affects recall more, with COV-R dropping to 49.38% and AMR-R rising to 0.7609, showing RECAP edges are key for coverage. The largest performance drop occurs when both BRICS and

Table 4: Quality of generated conformation ensembles for the GEOM-DRUGS test set in terms of Coverage (%) and Average Minimum RMSD (Å) with $\delta = 0.75$ Å with 5000 training samples.

| Models | COV-R (%) ↑ | | AMR-R (Å) ↓ | | COV-P (%) ↑ | | AMR-P (Å) ↓ | |
|---|---|---|---|---|---|---|---|---|
| | Mean | Median | Mean | Median | Mean | Median | Mean | Median |
| FADiff | **51.17** | **50.10** | **0.7519** | **0.7503** | **34.51** | **21.61** | **1.0389** | **1.0224** |
| w/o BRICS | 50.85 | 49.61 | 0.7568 | 0.7575 | 33.93 | 20.57 | 1.0461 | 1.0313 |
| w/o RECAP | 49.38 | 48.42 | 0.7609 | 0.7639 | 34.18 | 21.21 | 1.0420 | 1.0247 |
| w/o B & R | 48.60 | 46.89 | 0.7684 | 0.7708 | 33.74 | 19.86 | 1.0492 | 1.0377 |

RECAP edges are removed (w/o B & R), with COV-R at 48.60% and COV-P at 33.74%, highlight the complementary roles of BRICS and RECAP related bonds. These results underscore the potential of incorporating chemical semantic knowledge, such as BRICS and RECAP reaction edges, in enhancing chemical generative models, as both play crucial roles in generating diverse and accurate conformations.

### 4.3 REVERSE DIFFUSION STEPS

In Fig. 3, we vary the number of steps used in the reverse diffusion process and evaluate the performance on GEOM-DRUGS. It shows that torsional diffusion-based methods are all parsimonious in terms of number of steps required: the majority of gain in performance over prior diffusion-based methods is attained with only 10 steps. The FADiff models have demonstrated even higher efficiency, achieving superior performance with

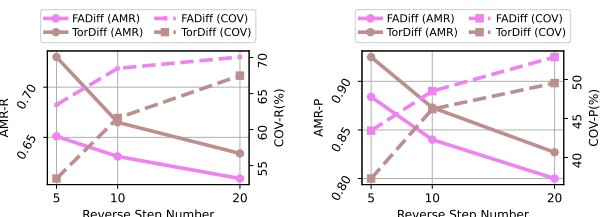

(a). Mean AMR-R & COV-R.  (b). Mean AMR-P & COV-P.

Figure 3: Reverse steps *v.s.* generation quality.

fewer steps compared to other methods. This highlights the enhanced sampling efficiency of FADiff, as it is able to generate high-quality conformers with significantly reduced computational cost. The ability to maintain strong performance with fewer diffusion steps underscores the effectiveness of the fragmentation-based approach in accelerating the reverse diffusion process.

### 4.4 FURTHER EXPERIMENTAL RESULTS

We provide further results and discussions on GEOM-QM9, GEOM-XL, minimum fragment size $z$, conformer matching, property prediction, and fragmentation methods in Appendix D.

## 5 CONCLUSION

In this work, we propose Fragment-Augmented Diffusion (FADiff), a novel framework for molecular conformation generation that incorporates molecular fragmentation as a data augmentation strategy within diffusion models. By using molecular fragmentation as a data augmentation strategy, FADiff effectively captures local structural variations while preserving the integrity of the entire molecule. Our extensive experiments demonstrate that FADiff consistently outperforms state-of-the-art methods in generating diverse and accurate conformations particularly in data-scarce scenarios where augmented data significantly enhances model performance. Additionally, we provided a comprehensive analysis of different fragmentation strategies and their impact on the model's effectiveness, offering valuable insights into how the inclusion of chemical rules influences generation quality. This work molecular conformation generation by enhancing the exploration of conformational space, with promising applications in areas such as drug discovery and materials science.

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

# A SUPPLEMENTARY THEROMS AND PROOFS

## A.1 DETAILED EXPLANATION OF TORSIONAL DIFFUSION

In the main text, we provided a high-level overview of the torsion diffusion basics. Here, we offer a more detailed explanation of the underlying technical aspects and mathematical formulations. The goal of conformer generation is to learn the probability distribution $p_\theta(\tau, \mathcal{G})$.

**Diffusion Modeling on the Hypertorus Space.** Since each torsion angle lies in the range $[0, 2\pi)$, the $m$ torsion angles of a conformer define a hypertorus $\mathbb{T}^m$ (Jing et al., 2022). To model the generative process over this space, we apply the continuous score-based framework of (Song & Ermon, 2020), which extends to data distributions on compact Riemannian manifolds, such as $\mathbb{T}^m$ (De Bortoli, 2022). Specifically, for a Riemannian manifold $M$, let $\mathbf{x} \in M$, let $\mathbf{w}$ be the Brownian motion on the manifold, and let the drift $\mathbf{f}(\mathbf{x}, t)$, score $\nabla_\mathbf{x} \log p_t(\mathbf{x})$, and score model output $\mathbf{s}(\mathbf{x}, t)$ be elements of the tangent space $T_\mathbf{x}M$. The reverse stochastic differential equation (SDE) on the manifold can be discretized and solved as a *geodesic random walk*, starting with samples from $p_T(\mathbf{x})$ to approximately recover the original data distribution $p_0(\mathbf{x})$ (De Bortoli, 2022).

**Noise Scale and Forward Diffusion.** For the forward diffusion process, we use rescaled Brownian motion, where the drift $\mathbf{f}(\mathbf{x}, t) = 0$ and the noise scale is given by $\mathbf{g}(t) = \sqrt{\frac{d}{dt}\sigma^2(t)}$. We adopt an exponential diffusion schedule $\sigma(t) = \sigma_{\min}e^{t \log \frac{\sigma_{\max}}{\sigma_{\min}}}$, as in Song & Ermon (2019); Ho et al. (2020); Song & Ermon (2020), with parameters $\sigma_{\min} = 0.01\pi$ and $\sigma_{\max} = \pi$, for $t \in (0, 1)$. Due to the compactness of the manifold, the prior distribution $p_T(\mathbf{x})$ is not Gaussian, but a uniform distribution over the manifold $M$.

**Handling Periodicity with Wrapped Normal Distributions.** To respect the periodic nature of torsion angles, we treat the torus $\mathbb{T}^m \cong [0, 2\pi)^m$ as the quotient space $\mathbb{R}^m/2\pi\mathbb{Z}^m$, where equivalence relations $(\tau_1, \ldots, \tau_m) \sim (\tau_1 + 2\pi, \ldots, \tau_m + 2\pi)$ hold (Jing et al., 2022). The perturbation kernel for rescaled Brownian motion on $\mathbb{T}^m$ is the *wrapped normal distribution* on $\mathbb{R}^m$. Specifically, for any $\tau, \tau' \in [0, 2\pi)^m$, the perturbation kernel is given by: $p_{t|0}(\tau' \mid \tau) \propto \sum_{d\in\mathbb{Z}^m} \exp\left(-\frac{\|\tau'-\tau+2\pi d\|^2}{2\sigma^2(t)}\right)$, where $\sigma(t)$ is the noise scale. We sample from this kernel by first sampling from the corresponding unwrapped isotropic normal distribution and then applying elementwise mod $2\pi$ to ensure periodicity. Finally, we can train the score model involves minimizing the denoising score matching (DSM) loss. The DSM loss function is defined as:

$$J_{\text{DSM}}(\theta) = \mathbb{E}_t\left[\lambda(t)\mathbb{E}_{\tau_0\sim p_0, \tau\sim p_{t|0}(\cdot|\tau_0,\mathcal{G})}\left[\|\mathbf{s}_\theta(C, t) - \nabla_\tau \log p_{t|0}(\tau \mid \tau_0, \mathcal{G})\|^2\right]\right],$$

where $\lambda(t)$ are precomputed weight factors that balance the contribution of different time steps. The tangent space $T_\tau\mathbb{T}^m$ is equivalent to $\mathbb{R}^m$, so all operations in the loss computation are straightforward.

**Dihedral Angle Calculation** In molecular torsional geometry, the *dihedral angle* describes the relative orientation of four atoms connected by three consecutive bonds (De Bortoli, 2022; Jing et al., 2022). For four atoms $a$, $b$, $c$, and $d$, the torsion angle $\tau_{abcd}$ is the angle between the plane formed by atoms $a$, $b$, and $c$ and the plane formed by atoms $b$, $c$, and $d$.

The torsion angle is calculated using the normal vectors of these planes. Let:

$$\mathbf{u}_{ab} = x_b - x_a, \quad \mathbf{u}_{bc} = x_c - x_b, \quad \mathbf{u}_{cd} = x_d - x_c$$

The normal vectors to the planes $abc$ and $bcd$ are:

$$\mathbf{n}_{abc} = \mathbf{u}_{ab} \times \mathbf{u}_{bc}, \quad \mathbf{n}_{bcd} = \mathbf{u}_{bc} \times \mathbf{u}_{cd}$$

The cosine and sine of the torsion angle $\tau_{abcd}$ are given by:

$$\cos \tau_{abcd} = \frac{\mathbf{n}_{abc} \cdot \mathbf{n}_{bcd}}{|\mathbf{n}_{abc}||\mathbf{n}_{bcd}|}$$

$$\sin \tau_{abcd} = \frac{\mathbf{u}_{bc} \cdot (\mathbf{n}_{abc} \times \mathbf{n}_{bcd})}{|\mathbf{u}_{bc}||\mathbf{n}_{abc}||\mathbf{n}_{bcd}|}$$

This gives the torsion angle $\tau_{abcd} \in [0, 2\pi)$, which describes the relative rotation of the two planes.

**Torsional Score Framework and Update.** Learning a score model $\mathbf{s}_\theta(C, t)$ over intrinsic coordinates is challenging due to the dependence of torsional space dimensionality $m$ on the molecular graph $\mathcal{G}$, and the influence of both $\mathcal{G}$ and local structures $L$ on the mapping to conformers. Additionally, torsion angles vary with arbitrary reference neighbors, adding ambiguity. A simpler approach represents a conformer $C \in \mathcal{C}_G$ using extrinsic (Cartesian) coordinates as a 3D point cloud modulo global roto-translation: $\mathcal{C}_G \cong \mathbb{R}^{3n}/SE(3)$. The score model $\mathbf{s}_\theta(C, t)$ is then defined over $\mathcal{C}_G$, predicting $SE(3)$-invariant scalar quantities for each bond, simplifying the learning process (De Bortoli, 2022).

Instead of explicitly defining each torsion angle $\tau_i$, Torsional Diffusion (Jing et al., 2022) leverages the fact that changing $\tau_i$ by $\Delta\tau_i$ can be applied directly to the 3D atomic coordinates. Geometrically, this corresponds to a relative rotation of atoms around the bond, applied directly in 3D space. This intuition is formalized as follows: Let $(b_i, c_i)$ be a rotatable bond, and let $\mathbf{x}_{v(b_i)}$ be the positions of atoms on the $b_i$-side of the molecule. Let $R(\theta, x_{c_i}) \in SE(3)$ be the rotation by Euler vector $\theta$ about $x_{c_i}$. Then for $C, C' \in \mathcal{C}_G$, if $\tau_i$ is any definition of the torsion angle around bond $(b_i, c_i)$, we have:

$$\tau_i(C') = \tau_i(C) + \theta \qquad \text{if} \quad \exists x \in C, x' \in C', \quad \mathbf{x}'_{v(b_i)} = \mathbf{x}_{v(b_i)},$$

$$\tau_j(C') = \tau_j(C) \qquad\qquad \forall j \neq i, \quad \mathbf{x}'_{v(c_i)} = R(\theta \hat{\mathbf{b}}_{c_i}, x_{c_i}) \mathbf{x}_{v(c_i)}$$

where $\hat{\mathbf{b}}_{c_i} = \frac{x_{c_i} - x_{b_i}}{\|x_{c_i} - x_{b_i}\|}$.

The core idea of the proof is to show that rotating the bond $(b_i, c_i)$ by $\theta$ changes only the torsion angle $\tau_i$, while leaving the torsion angles $\tau_j$ for all other bonds $j \neq i$ unchanged. This is achieved by applying a rotation centered at $x_{c_i}$, which affects only the atoms on the $c_i$-side of the molecule. The Rodrigues rotation formula demonstrates that the relative positions of atoms on the $b_i$-side remain fixed, while the atoms on the $c_i$-side undergo a rotation by $\theta$, resulting in the desired change in $\tau_i$. A full proof can be found in (Jing et al., 2022). To apply a torsion update $\Delta\tau = (\Delta\tau_1, \ldots, \Delta\tau_m)$, the updates $\Delta\tau_i$ are applied sequentially in any order. Since training and inference only use torsion updates $\Delta\tau$, this approach operates solely on 3D point clouds and the updates applied to them. Local structures $L$ can be generated from RDKit by producing full 3D conformers $C \in \mathcal{C}_G$ and randomizing all torsion angles to sample uniformly over $\mathbb{T}^m$. Torsion updates are predicted directly from, and applied to, the 3D point cloud, eliminating the need for selecting reference neighbors for any $\tau_i$, thus ensuring invariance to such choices.

**Equivariance.** The torsional score model must be $SE(3)$-invariant, but also respects an additional symmetry: physical energy is invariant (or nearly so) under parity inversion, which is essential in machine learning for atomic systems, ensuring that vectors of atomic dipoles or forces rotate according to the conformation coordinates (Quack, 2002; Xu et al., 2022). Thus, integrating such inductive bias into model parameterization for 3D geometry is crucial for generalization (Quack, 2002; Xu et al., 2022).

This requires the learned density to satisfy $p(C) = p(-C)$, where $-C = \{-x \mid x \in C\}$. For the conditional distribution over torsion angles, this implies $p(\tau(C) \mid L(C)) = p(\tau(-C) \mid L(-C))$. Consequently, for all diffusion times $t$,

$$\nabla_\tau \log p_t(\tau(C) \mid L(C)) = -\nabla_\tau \log p_t(\tau(-C) \mid L(-C))$$

Since the score model learns $\mathbf{s}(C, t) = \nabla_\tau \log p_t(\tau(C) \mid L(C))$, it follows that $\mathbf{s}(C, t) = -s_G(-C, t)$. Therefore, the score model must be $SE(3)$-invariant but equivariant (change sign) under parity inversion, outputting pseudoscalars in $\mathbb{R}^m$.

## A.2 SCORE NETWORK ARCHITECTURE

To predict torsion scores under $SE(3)$-invariant and parity-equivariant constraints, we follow the framework used in Torsional Diffusion, which contains three main components: an embedding layer, $K$ interaction layers, and a pseudotorque layer. The pseudotorque layer outputs pseudoscalar torsion scores $\Delta\tau := \partial \log p/\partial\tau$ for each rotatable bond.

1. In the embedding layer, we construct a radius graph on top of the molecular graph, generating initial scalar embeddings for nodes and edges. Node embeddings $V_a^{(0,0,1)}$ combine chemical features and sinusoidal time embeddings, while edge embeddings $e_{ab}$ incorporate bond distances using radial basis functions and chemical features. This setup ensures that both local atomic environments and temporal information are captured.

2. The interaction layers are built using E(3)NN convolutional layers, which propagate messages between nodes by combining irreducible representations of node features with spherical harmonics of the normalized edge vectors (Geiger & Smidt, 2022). These messages are aggregated using Clebsch-Gordan coefficients, ensuring that the node representations remain $SE(3)$-equivariant. At each layer, the interaction is governed by tensor products of node and edge features, and the rotational order of the representations is restricted to be at most 2 (Thomas et al., 2018).

3. The pseudotorque layer predicts torsion scores by constructing tensor-valued filters centered on each rotatable bond. These filters are formed from the tensor product of spherical harmonics with a $l = 2$ representation of the bond axis. The convolution with neighboring node features produces pseudoscalar outputs, which are passed through odd-function dense layers (e.g., with tanh nonlinearity) to generate the final torsion score predictions. This layer is inspired by the concept of torque, ensuring the correct symmetry properties for torsion score prediction.

The complete architecture design and tensor computation pipeline can be found in (Jing et al., 2022).

## A.3 EUCLIDEAN LIKELIHOOD CONVERSION

Torsional Diffusion framework computes the likelihood of torsional angles in the torsional space $p_\theta(\tau \mid L), \tau \in \mathbb{T}^m$. However, for compatibility with physical models which operate in Euclidean space, it is necessary to convert this torsional likelihood into a Euclidean likelihood $p(x \mid L), x \in \mathbb{R}^{3n}$. This conversion ensures that our model can be integrated with standard molecular simulation frameworks that rely on Euclidean coordinates. To achieve this, we introduce a conversion factor that accounts for the difference in volume elements between the torsional and Euclidean spaces. This factor is derived through the following relationship:

$$p_\theta(x \mid L) = \frac{p_\theta(\tau \mid L)}{8\pi^2 \sqrt{\det g}} \quad \text{where} \quad g_{\alpha\beta} = \sum_{k=1}^n J_\alpha^{(k)} \cdot J_\beta^{(k)}.$$

Here, the matrix $g_{\alpha\beta}$ represents the metric tensor that captures the relationship between the torsional angles and the Euclidean coordinates. The indices $\alpha, \beta$ range from 1 to $m + 3$, with $m$ representing the number of torsional degrees of freedom and the additional 3 accounting for global rotations.

The proof of this relationship involves constructing a manifold $M$ that represents the set of all centered conformers with fixed local structures but arbitrary torsion angles and orientations. The coordinates of this manifold include both the torsional angles and the global rotational degrees of freedom. By analyzing the covariant basis vectors of this manifold and computing the corresponding metric tensor, we derive the conversion factor between the volume elements in torsional and Euclidean spaces. The full proof, including the detailed derivation of the metric tensor and the integration over global rotations, can be found in Jing et al. (2022).

This conversion is crucial because it allows us to bridge the gap between torsional space, where the likelihood is naturally defined, and Euclidean space, where physical simulations and energy-based models operate. By ensuring that our likelihoods are compatible with the Boltzmann measure, we can accurately model molecular systems and integrate our framework with existing simulation tools.

### A.4 DERIVATION OF THE ELBO

To derive the ELBO for Torsional Diffusion, we start by considering the parameterization of the reverse process. The reverse process is defined as:

$$\mu_\theta(\tau^t, t) = \frac{1}{\sqrt{\alpha_t}} \left( \tau^t - \frac{\beta_t}{\sqrt{1 - \bar{\alpha}_t}} \epsilon_\theta(\mathcal{G}, \tau^t) \right)$$

where $\epsilon_\theta$ is a neural network that predicts the noise necessary to correct the torsional angles $\tau^t$. The ELBO objective is given by:

$$L_{\text{ELBO}} = \sum_{t=1}^{T} \gamma_t \mathbb{E}_{\{\tau^0, \mathcal{G}\} \sim q(\tau^0, \mathcal{G}), \epsilon \sim \mathcal{N}(0, I)} \left[ \left\| \epsilon - \epsilon_\theta(\mathcal{G}, \tau^t) \right\|_2^2 \right]$$

where $\tau^t = \sqrt{\bar{\alpha}_t} \tau^0 + \sqrt{1 - \bar{\alpha}_t} \epsilon$, and $\gamma_t = \frac{\beta_t}{2\alpha_t(1 - \bar{\alpha}_{t-1})}$. We derive the ELBO by considering the expected log-likelihood:

$$\mathbb{E} \log p_\theta(\tau^0 | \mathcal{G}) = \mathbb{E} \log \mathbb{E}_{q(\tau^{1:T} | \tau^0)} \left[ \frac{p_\theta(\tau^{0:T-1} | \mathcal{G}, \tau^T) \times p(\tau^T)}{q(\tau^{1:T} | \tau^0)} \right]$$

Applying Jensen's inequality, we have:

$$\geq \mathbb{E}_q \log \left[ \frac{p_\theta(\tau^{0:T-1} | \mathcal{G}, \tau^T) \times p(\tau^T)}{q(\tau^{1:T} | \tau^0)} \right]$$

This can be expanded as:

$$= \mathbb{E}_q \left[ \log p(\tau^T) - \sum_{t=1}^{T} \log \frac{p_\theta(\tau^{t-1} | \mathcal{G}, \tau^t)}{q(\tau^t | \tau^{t-1})} \right]$$

Further simplification gives:

$$= \mathbb{E}_q \left[ \log \frac{p(\tau^T)}{q(\tau^T | \tau^0)} - \log p_\theta(\tau^0 | \mathcal{G}, \tau^1) - \sum_{t=2}^{T} \log \frac{p_\theta(\tau^{t-1} | \mathcal{G}, \tau^t)}{q(\tau^t | \tau^0)} \right]$$

The ELBO is then expressed as:

$$L_{\text{ELBO}} = -\mathbb{E}_q \left[ \text{KL} \left( q(\tau^T | \tau^0) \| p(\tau^T) \right) + \sum_{t=1}^{T-1} \text{KL} \left( q(\tau^{t-1} | \tau^t, \tau^0) \| p_\theta(\tau^{t-1} | \mathcal{G}, \tau^t) \right) \right].$$

The KL divergence is calculated based on the Gaussian distributions $q$ and $p_\theta$, sharing the same covariance matrix $\bar{\beta}_t I$.

### A.5 LEMMA 1.

The lemma is a direct consequence of the results in (Poole et al., 2019). It shows that minimizing the objective function for fragmentation-based torsion modeling is equivalent to maximizing the mutual information between the torsion angles of the full molecular structure $\{\hat{\boldsymbol{\tau}}_b\}_{b=1}^{B+1}$ and those true torsion angles of the fragment $\{\boldsymbol{\tau}_b\}_{b=1}^{B+1}$. By focusing on valid torsion subspaces, this approach ensures that fragments retain relevant torsional and geometric information from $\hat{\boldsymbol{\tau}}_b$. Maximizing the mutual information between $\hat{\boldsymbol{\tau}}_b$ and $\boldsymbol{\tau}_b$ guarantees that local fragment optimizations contribute to a globally consistent molecular conformation, particularly in terms of torsional flexibility and stability. As a result, the optimal fragmentation $h^*$ preserves the maximum possible information, ensuring that the fragmented representation retains the essential characteristics of the original structure, leading to improved model performance by aligning the fragmented and original structures as closely as possible.

# B  FRAGMENTATION-BASED DECOMPOSITION AND ERROR ANALYSIS

## B.1  FRAGMENTATION RULES

To validate the effectiveness and differences of various decomposition methods in learning conformation structures in the fragmentation torsion space, we analyzed several fragmentation rules in FADiff implementations. These methods provide domain knowledge and deeper insights into the task. The analyzed rules used include the BRICS method and RECAPS method, and a graph-based fragmentation method can be employed by analyzing the connectivity of molecular graphs, which is also the method that Torsional Diffusion used for selecting the rotatable bond (Gordon et al., 2012; Stern et al., 2020; Jing et al., 2022), this method identifies cut edges by examining whether the removal of an edge disconnects the graph into separate components.

**RECAP (Retrosynthetic Combinatorial Analysis Procedure)**  (Lewell et al., 1998)

RECAP is a classical technique aimed at decomposing complex molecules into smaller, manageable fragments through retrosynthetic analysis. The core principle involves identifying and cleaving chemical bonds that are common in organic synthesis, such as ester, amide, and ether bonds. These bonds are selected based on their prevalence and the ease of cleavage, prioritizing those that connect functional groups to generate fragments with clear chemical functionalities. Typically, RECAP employs a single-cut strategy, focusing on one bond at a time, resulting in basic fragments. This simplicity allows for rapid generation of fragments suitable for combinatorial chemistry and initial drug screening.

**BRICS (Breaking of Retrosynthetically Interesting Chemical Substructures)**  (Degen et al., 2008)

BRICS improves upon RECAP by offering a more detailed approach to fragment generation. It applies a comprehensive set of rules to identify and cleave key substructures in chemical compounds, considering not just bond types but also the surrounding chemical environment, such as aromaticity and heterocycles. This allows BRICS to generate more complex and diverse fragments, supporting multi-functional group cleavage to produce synthetically feasible and biologically relevant fragments. BRICS focuses on creating diverse, drug-like fragments, making it valuable for drug design. Researchers can use BRICS to build flexible and accurate fragment libraries for virtual screening and molecular optimization. While RECAP is suitable for basic fragment analysis, BRICS provides a more advanced tool for high-precision drug development and compound optimization.

**Graph-based Fragmentation**  (Gordon et al., 2012; Stern et al., 2020; Jing et al., 2022) We consider a bond freely rotatable if severing the bond creates two connected components of the total graph, each of which has at least two atoms. It guarantees that torsion angles in cycles (or rings), which cannot be rotated independently, are considered part of the local structure. It can be described as following steps:

- Convert the molecular graph into an undirected graph $\mathcal{G}$.
- For each edge, temporarily remove it and check if the resulting graph remains connected.
- If the graph becomes disconnected, identify the connected components and classify them as fragments.
- Store the edges whose removal results in disconnected components, as these represent potential cut points for fragmentation.

Thus, it allows for the identification of edges that, when removed, split the molecular graph into meaningful substructures. The algorithm ensures that fragments retain their connectivity, making it particularly useful for identifying torsion-related substructures.

## B.2  FRAGMENTATION AUGMENTATION ERROR ANALYSIS

During molecular fragmentation, chemical or graph-based cut rules decompose the molecule into smaller fragments. The choice of cut edges introduces errors between the torsion angles $\hat{\tau}_b^{u,v}$ from

the full molecular graph and the true torsion angles $\tau_b^{u,v}$ of the fragments due to potential loss of structural or torsional information. To quantify this, we define an error term $\epsilon$ capturing the deviation:

$$\hat{\tau}_b^{u,v} = \tau_b^{u,v} + \epsilon.$$

This error $\epsilon$ depends on factors like cut edge selection and preservation of the local chemical environment. To understand its impact on the mutual information and model performance, we derive a general error bound and analyze how different fragmentation methods affect this bound.

The mutual information between $\hat{\tau}_b^{u,v}$ and $\tau_b^{u,v}$ is given by:

$$I(\hat{\tau}_b^{u,v}; \tau_b^{u,v}) = H(\tau_b^{u,v}) - H(\tau_b^{u,v} \mid \hat{\tau}_b^{u,v}),$$

where $H(\tau_b^{u,v})$ is the entropy of the fragment torsional angles, and $H(\tau_b^{u,v} \mid \hat{\tau}_b^{u,v})$ is the conditional entropy of $\tau_b^{u,v}$ given $\hat{\tau}_b^{u,v}$. Our goal is to understand how the error $\epsilon$ influences this mutual information and, consequently, the accuracy of our molecular modeling.

Assuming that the error $\epsilon$ is a random variable with a probability density function $p(\epsilon)$, we can express the conditional entropy $H(\tau_b^{u,v} \mid \hat{\tau}_b^{u,v})$ in terms of the entropy of the error $\epsilon$:

$$H(\tau_b^{u,v} \mid \hat{\tau}_b^{u,v}) = h(\epsilon),$$

where $h(\epsilon)$ denotes the differential entropy of $\epsilon$. This is because, given $\hat{\boldsymbol{\tau}} = \boldsymbol{\tau} + \epsilon$, the uncertainty in $\boldsymbol{\tau}$ given $\hat{\boldsymbol{\tau}}$ is entirely due to the uncertainty in $\epsilon$. If we model the error $\epsilon$ as a zero-mean Gaussian random variable with variance $\sigma^2$ (i.e., $\epsilon \sim \mathcal{N}(0, \sigma^2)$), its differential entropy is:

$h(\epsilon) = -\int_{-\infty}^{\infty} f_\epsilon(\epsilon) \ln f_\epsilon(\epsilon) \, d\epsilon$, where $f_\epsilon(\epsilon) = \frac{1}{\sqrt{2\pi\sigma^2}} \exp\left(-\frac{\epsilon^2}{2\sigma^2}\right)$.

$$
\begin{aligned}
h(\epsilon) &= -\int_{-\infty}^{\infty} f_\epsilon(\epsilon) \left(\ln f_\epsilon(\epsilon)\right) d\epsilon = -\int_{-\infty}^{\infty} f_\epsilon(\epsilon) \left(-\frac{1}{2}\ln(2\pi\sigma^2) - \frac{\epsilon^2}{2\sigma^2}\right) d\epsilon \\
&= \int_{-\infty}^{\infty} f_\epsilon(\epsilon) \left(\frac{1}{2}\ln(2\pi\sigma^2) + \frac{\epsilon^2}{2\sigma^2}\right) d\epsilon = \frac{1}{2}\ln(2\pi\sigma^2) \int_{-\infty}^{\infty} f_\epsilon(\epsilon) \, d\epsilon + \frac{1}{2\sigma^2} \int_{-\infty}^{\infty} \epsilon^2 f_\epsilon(\epsilon) \, d\epsilon \\
&= \frac{1}{2}\ln(2\pi\sigma^2) \cdot 1 + \frac{1}{2\sigma^2} \cdot \mathbb{E}[\epsilon^2] = \frac{1}{2}\ln(2\pi\sigma^2) + \frac{1}{2\sigma^2} \cdot \sigma^2 \\
&= \frac{1}{2}\ln(2\pi\sigma^2) + \frac{1}{2} = \frac{1}{2}\left(\ln(2\pi\sigma^2) + 1\right) \\
&= \frac{1}{2}\ln(2\pi e \sigma^2),
\end{aligned}
$$

and we have:

$$I(\hat{\tau}_b^{u,v}; \tau_b^{u,v}) = H(\tau_b^{u,v}) - h(\epsilon) = H(\tau_b^{u,v}) - \frac{1}{2}\ln(2\pi e \sigma^2).$$

This equation reveals that the mutual information decreases as the error variance $\sigma^2$ increases.

### B.3  IMPACT OF FRAGMENTATION RULES ON THE ERROR BOUND

Different fragmentation rules influence the error variance $\sigma^2$ (or mean squared error, MSE) between the torsion angles of the fragments $\tau_b^{u,v}$ and those of the full molecule $\hat{\tau}_b^{u,v}$, thus affecting the mutual information $I(\tau_b^{\hat{u},v}; \tau_b^{u,v})$ between them. A lower error variance implies a stronger relationship between $\tau_b^{\hat{u},v}$ and $\tau_b^{u,v}$, leading to enhanced model accuracy. Below, we analyze how different fragmentation rules impact the error variance $\sigma^2$ and discuss strategies to minimize it.

### B.4  FACTORS AFFECTING ERROR VARIANCE $\sigma^2$

To minimize the error variance $\sigma^2$ and maximize the mutual information $I(\hat{\boldsymbol{\tau}}; \boldsymbol{\tau})$ between the torsional angles of the fragments $\boldsymbol{\tau}$ and those of the full molecule $\hat{\boldsymbol{\tau}}$, fragmentation methods should carefully consider several key factors. Each factor influences the torsional properties by affecting the torsional potential energy surfaces and the distributions of torsion angles (Stern et al., 2020; Horton et al., 2022; Stern et al., 2022). Below, we provide a detailed examination of each factor from an *energy perspective*, including examples and mathematical definitions where applicable.

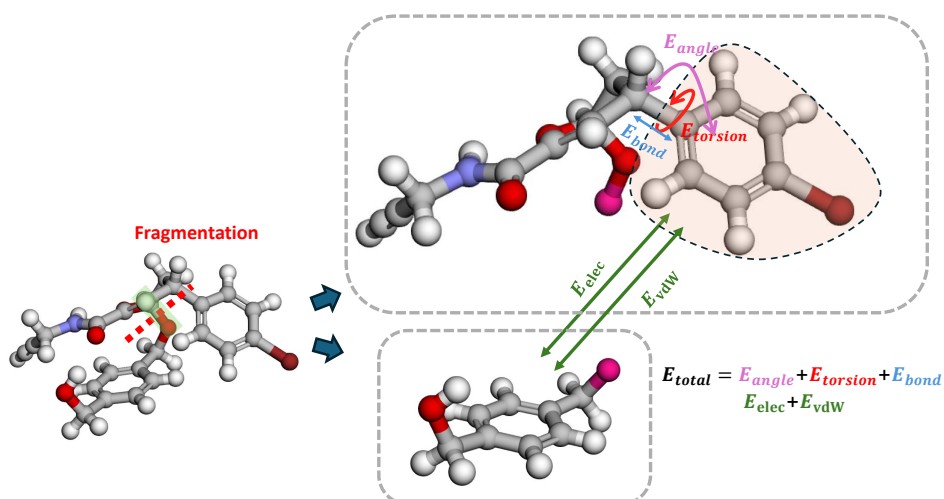

Figure 4: Visualization of energy-based analysis.

### B.4.1 ENERGY PERSPECTIVE

The force field (FF) in molecular simulations describes the potential conformational energy $E_{\text{conf}}$ of a system of atoms or molecules. It comprises mathematical expressions and associated parameters to model both bonded and non-bonded interactions. The total conformational energy of the system is given by (Kang et al., 1996; Kania et al., 2021):

$$E_{\text{conf}} = E_{\text{bonded}} + E_{\text{nonbonded}} \tag{2}$$

**Bonded Interactions**   Bonded interactions account for atoms connected by chemical bonds and include bond stretching, angle bending, and dihedral (torsional) rotations:

$$E_{\text{bonded}} = \sum_{\text{bonds}} k_b(b - b_0)^2 + \sum_{\text{angles}} k_\theta(\theta - \theta_0)^2 + \sum_{\text{dihedrals}} \sum_n \frac{V_n}{2}[1 + \cos(n\phi - \gamma_n)] \tag{3}$$

- **Bond Stretching:** Harmonic potential where $k_b$ is the bond force constant, $b$ is the bond length, and $b_0$ is the equilibrium bond length.

- **Angle Bending:** Harmonic potential where $k_\theta$ is the angle force constant, $\theta$ is the bond angle, and $\theta_0$ is the equilibrium bond angle.

- **Dihedral (Torsional) Rotation:** Fourier series expansion where $V_n$ is the torsional barrier amplitude, $n$ is the periodicity, $\phi$ is the dihedral angle, and $\gamma_n$ is the phase offset.

Alternatively, the bonded energy term can be rewritten as torsional energy:

$$E_{\text{torsion}} = \sum_i \sum_n K_n^{(i)}[1 + \cos(n\phi_i - \gamma_n^{(i)})] \tag{4}$$

where $K_n^{(i)}$ and $\gamma_n^{(i)}$ are the torsional parameters specific to dihedral $i$, and the summation over $n$ typically includes terms up to $n = 4$ by most of the packages applied in molecular dynamics simulations (Kania et al., 2021).

**Non-bonded Interactions**   Non-bonded interactions consider pairs of atoms not directly bonded and include Van der Waals forces and electrostatic interactions:

$$E_{\text{nonbonded}} = \sum_{i<j} \left[ 4\varepsilon_{ij} \left( \left( \frac{\sigma_{ij}}{r_{ij}} \right)^{12} - \left( \frac{\sigma_{ij}}{r_{ij}} \right)^{6} \right) + \frac{q_i q_j}{4\pi\varepsilon_0 \varepsilon_r r_{ij}} \right] \tag{5}$$

- **Van der Waals (Lennard-Jones) Potential:** Described by parameters $\varepsilon_{ij}$ (depth of the potential well) and $\sigma_{ij}$ (finite distance at which the interparticle potential is zero), with $r_{ij}$ being the distance between atoms $i$ and $j$.
- **Electrostatic Potential:** Coulombic interaction where $q_i$ and $q_j$ are the partial charges of atoms $i$ and $j$, $\varepsilon_0$ is the vacuum permittivity, and $\varepsilon_r$ is the relative permittivity (dielectric constant).

To analyze how different fragmentation methods influence the error variance $\sigma^2$ and, consequently, the estimation effectiveness in fragment-based molecular modeling, we consider the molecular conformational energy from an energy perspective. The conformational energy $E_{\text{conf}}$ of a molecule is given by:

$$E_{\text{conf}} = \sum_{i} \sum_{n} K_n^{(i)} \left[ 1 + \cos\left( n\phi_i - \gamma_n^{(i)} \right) \right] + \sum_{i<j} \left[ 4\varepsilon_{ij} \left( \left( \frac{\sigma_{ij}}{r_{ij}} \right)^{12} - \left( \frac{\sigma_{ij}}{r_{ij}} \right)^{6} \right) + \frac{q_i q_j}{4\pi\varepsilon_0 \varepsilon_r r_{ij}} \right]. \tag{6}$$

The first term represents the torsional (dihedral) interactions, where $K_n^{(i)}$ and $\gamma_n^{(i)}$ are the torsional parameters for dihedral $i$, $\phi_i$ is the dihedral angle, and the sum over $n$ includes the relevant periodicities. The second term accounts for non-bonded interactions, including Van der Waals forces modeled by the Lennard-Jones potential and electrostatic interactions modeled by Coulomb's law, with $\varepsilon_{ij}, \sigma_{ij}, q_i$, and $q_j$ being the Van der Waals parameters and partial charges, respectively (Kania et al., 2021).

Fragmentation methods impact $E_{\text{conf}}$ by altering both bonded and non-bonded interactions. These alterations affect the torsional potential energy surfaces and, consequently, the torsion angle distributions $\boldsymbol{\tau}$, leading to variations in the error variance $\sigma^2$. We examine how different fragmentation strategies affect the terms in Equation equation 6 and discuss their implications for the estimation effectiveness.

### B.4.2 Effect of Fragmentation on Torsional Energy

Fragmentation can significantly impact the torsional energy terms in the conformational energy expression. When fragments are created by cutting bonds, the parameters $K_n^{(i)}$ and $\gamma_n^{(i)}$ associated with the torsional angles $\phi_i$ may change due to the alteration of the local chemical environment. This is especially pertinent in the following scenarios:

**Disruption of Conjugation and Resonance.** Fragmenting through bonds that are part of conjugated systems or aromatic rings disrupts electron delocalization. This alteration affects the torsional barrier heights $K_n^{(i)}$ and phase offsets $\gamma_n^{(i)}$, modifying the torsional potential energy surface and leading to discrepancies between the torsional angles in the fragment $\boldsymbol{\tau}$ and those in the full molecule $\hat{\boldsymbol{\tau}}$. The disruption can be quantified by changes in electron density distributions $\rho_{\text{frag}}(\mathbf{r})$ versus $\rho_{\text{full}}(\mathbf{r})$, impacting the energy landscape.

**Loss of Steric Interactions.** Removing bulky substituents adjacent to torsional bonds reduces steric hindrance, altering the energy landscape. The decrease in steric interactions can lower torsional barriers and shift equilibrium angles, causing differences between $\boldsymbol{\tau}$ and $\hat{\boldsymbol{\tau}}$. This effect can be modeled using steric energy terms in force fields, such as Van der Waals interactions, which are sensitive to atomic radii and distances.

### B.4.3 Effect of Fragmentation on Non-bonded Interactions

Fragmentation alters non-bonded interactions, which play a critical role in determining conformational preferences. The fragmentation process affects Van der Waals and electrostatic interactions as follows:

**Modification of Van der Waals Interactions.** By removing atoms and groups during fragmentation, the number of atom pairs contributing to the Van der Waals interactions decreases. Interactions spanning the fragmentation site are lost, and the balance of attractive and repulsive forces changes, affecting the conformational energy surface. The Van der Waals energy, dependent on parameters $\varepsilon_{ij}$ and $\sigma_{ij}$, is sensitive to changes in atomic pairs $i < j$.

**Changes in Electrostatic Interactions.** Fragmentation can delete or modify charged or polar groups, altering the distribution of partial charges $q_i$ and $q_j$. Disruption of hydrogen bonds and other electrostatic interactions modifies the energy landscape, potentially leading to different conformations in the fragments compared to the full molecule. The Coulombic potential is directly affected by the presence or absence of charged species and their spatial arrangement.

### B.4.4 IMPLICATIONS FOR FRAGMENTATION EFFECTIVENESS

The changes in torsional and non-bonded energy terms resulting from fragmentation have significant implications for the effectiveness of fragmentation methods in fragment-based molecular modeling. A higher error variance $\sigma^2$ indicates a weaker correspondence between the fragment torsional angles $\tau_b^{u,v}$ and those of the full molecule $\hat{\tau}_b^{u,v}$, as quantified by the mutual information:

$$I(\hat{\tau}_b^{u,v}; \tau_b^{u,v}) = H(\tau_b^{u,v}) - \frac{1}{2} \ln \left(2\pi e\, \sigma^2\right). \tag{7}$$

To minimize $\sigma^2$, effective fragmentation methods should aim to preserve the key energy terms in Equation equation 6. This involves:

**Preserving Electronic Effects.** Avoiding fragmentation through conjugated systems or aromatic rings maintains the torsional parameters $K_n^{(i)}$ and $\gamma_n^{(i)}$. By preserving electron delocalization, the torsional energy surfaces of the fragments remain similar to those of the full molecule, reducing deviations $\epsilon$.

**Retaining Steric Interactions.** Including bulky substituents and sterically significant groups in the fragments maintains steric hindrance, preserving torsional barriers and equilibrium angles. Quantitatively, this ensures that the steric energy contributions, such as those from the Lennard-Jones potential, remain consistent between the fragment and the full molecule.

**Maintaining Non-bonded Interactions.** Preserving key non-bonded interactions—especially hydrogen bonds and electrostatic attractions—helps maintain conformational preferences influenced by these forces. Retaining charged or polar groups ensures that the electrostatic interactions in the fragment mirror those in the full molecule.

Thus, by using fragmentation methods that preserve key electronic effects, steric interactions, and non-bonded interactions, we can reduce deviations $\epsilon$, minimize the error variance $\sigma^2$, and enhance the mutual information $I(\hat{\tau}_b^{u,v}; \tau_b^{u,v})$ between fragments and the full molecule. This leads to more accurate and reliable fragment-based molecular models, improving tasks such as molecular conformation prediction and property estimation. By selecting fragmentation strategies that minimize alterations to the energy terms in Equation equation 6, we align with the optimal fragmentation strategy $\zeta^*$ that maximizes mutual information and minimizes the error bound.

### B.4.5 ANALYSIS OF FRAGMENTATION METHODS

In this section, we analyze three fragmentation methods—RECAP, BRICS, and the graph-based fragmentation method—from the energy perspective discussed earlier. We examine how each method ensures effectiveness by preserving key electronic effects, steric interactions, and non-bonded interactions, thereby impacting the error variance $\sigma^2$ and the mutual information $I(\hat{\boldsymbol{\tau}}; \boldsymbol{\tau})$. The following paragraphs provide detailed analyses of these methods.

**RECAP Fragmentation Method Analysis** The RECAP (Retrosynthetic Combinatorial Analysis Procedure) method provides a straightforward and efficient way to decompose molecules by cleaving them at common synthetic bonds such as esters, amides, and ethers Lewell et al. (1998). This simple approach effectively targets functional groups and preserves core structures, making it

valuable for generating synthetically accessible fragments. By focusing on bonds commonly manipulated in organic synthesis, RECAP facilitates the exploration of potential synthetic pathways and the identification of key structural components. For example, fragmenting acetaminophen at the amide bond between the phenol ring and the acetamide group yields *p*-aminophenol and an acetyl group, which are both significant intermediates in chemical synthesis.

**BRICS Fragmentation Method Analysis**  BRICS (Breaking of Retrosynthetically Interesting Chemical Substructures) generates synthetically feasible and biologically relevant fragments by applying detailed rules that consider both bond types and their chemical environments Degen et al. (2008). By avoiding cuts within conjugated systems and aromatic rings, it preserves critical structural features, maintaining consistent torsional parameters $K_n^{(i)}$ and $\gamma_n^{(i)}$ and thus reducing error variance. BRICS also retains bulky substituents and sterically significant groups, which is essential for accurate torsional barriers and equilibrium angles. By considering the chemical context at fragmentation sites, it preserves key non-bonded interactions like hydrogen bonds and electrostatic attractions, enhancing the electrostatic components of $E_{\text{conf}}$ and increasing mutual information $I(\hat{\boldsymbol{\tau}}; \boldsymbol{\tau})$. For example, fragmenting ibuprofen at the carboxylic acid linkage keeps the aromatic ring and isobutyl group intact, preserving important steric and electronic properties.

**Graph-Based Fragmentation Method Analysis**  Graph-based fragmentation focuses on the molecule's connectivity, identifying fragmentation points that yield meaningful substructures without disrupting critical bonds Gordon et al. (2012); Stern et al. (2020). By avoiding breaks in bonds essential for conjugation and resonance, it maintains consistent torsional parameters $K_n^{(i)}$ and $\gamma_n^{(i)}$, thereby reducing error variance. This approach retains steric interactions by keeping bulky groups connected, which is crucial for accurate modeling of torsional barriers and equilibrium angles. Additionally, it preserves non-bonded interactions such as hydrogen bonds and electrostatic attractions by maintaining the connectivity of functional groups, enhancing mutual information $I(\hat{\boldsymbol{\tau}}; \boldsymbol{\tau})$. For instance, when fragmenting benzene, preserving the aromatic ring maintains its unique electronic properties and associated torsional parameters, whereas breaking bonds within the ring would eliminate these characteristics.

**Bridging the Analysis with Experimental Results**  The experimental results in Table 4 support our theoretical analysis of fragmentation methods' impact on error variance ($\sigma^2$) and mutual information ($I(\hat{\boldsymbol{\tau}}; \boldsymbol{\tau})$). The full FADiff model, incorporating the graph-based fragmentation method with both BRICS and RECAP edges, achieves the best performance across all metrics, with the highest mean COV-R (51.17%) and COV-P (50.10%) and the lowest AMR-R and AMR-P values. Removing BRICS edges (w/o BRICS) decreases precision metrics—COV-P drops to 34.51% and AMR-P increases to 1.0461 Å—indicating that BRICS fragmentation is crucial for achieving high precision in conformer generation. Similarly, removing RECAP edges (w/o RECAP) adversely affects recall metrics, with COV-R decreasing to 49.38% and AMR-R increasing to 0.7609 Å, highlighting RECAP fragmentation's importance for comprehensive coverage of the conformational space. The most significant performance decline occurs when both BRICS and RECAP edges are removed (w/o B & R), resulting in the lowest COV-R (48.60%) and COV-P (46.89%) and the highest AMR-R and AMR-P values. This underscores the complementary roles of BRICS and RECAP in preserving critical aspects of molecular structure essential for accurate conformer generation. These findings confirm our theoretical framework that optimal fragmentation strategies ($\zeta^*$) maximizing mutual information and minimizing error variance enhance model performance. Selecting fragmentation strategies that align with theoretical principles is crucial for optimizing model performance in practical applications.

## C  REPRODUCBILITY

**Experimental Details**  For conformer ensemble generation on GEOM-DRUGS, we mainly followed the setup used in (Jing et al., 2022). We trained the Torsional Diffusion models on NVIDIA RTX A100 GPUs for 250 epochs using the Adam optimizer for GEOM-DRUGS and GEOM-QM9. The primary hyperparameters were optimized using the validation set, resulting in the following configurations: an initial learning rate of 0.001, a learning rate scheduler with a patience of 20 epochs, 4 network layers, a second-order maximum representation, a cutoff radius $r_{\text{max}}$ of 5 Å, and

the inclusion of batch normalization. Specifically, followed the setup used in (Jing et al., 2022), we use the model trained from GEOM-DRUGS for GEOM-XL evaluation. The results reported for FADiff utilize 20 reverse diffusion steps, consistent with the approach in Jing et al. (2022). The minimum fragment size $z$ was set to 10 for both GEOM-DRUGS and GEOM-XL, while no such limit was applied in the GEOM-QM9 experiments. The maximum fragmentation edge number $\kappa$ is set to 5 for all datasets. For boltzmann generation experiments, we trained the Torsional Diffusion model on NVIDIA RTX A100 GPUs for 250 epochs using the Adam optimizer. The hyperparameters were set as follows: initial learning rate of 0.001, learning rate scheduler patience of 20, 4 layers, 2nd order maximum representation, and batch normalization enabled.

Consistent with the approach in (Jing et al., 2022), for each molecule that has $K$ ground truth conformations, we generate 2000 conformations. The datasets were randomly divided into training, validation, and test sets with sizes as follows: for GEOM-DRUGS, there are 243,473 training samples, 30,433 validation samples, and 1,000 test samples; for GEOM-QM9, there are 106,586 training samples, 13,323 validation samples, and 1,000 test samples. Since GEOM-XL is used solely for testing, its test set includes all 102 molecules from the MoleculeNet dataset that contain at least 100 atoms.

**Local Structure Initialization** As it stated in (Jing et al., 2022), the set of possible stable local structures $L$ for a given molecule is highly constrained and can be accurately predicted using fast cheminformatics methods, such as RDKit ETKDG (Riniker & Landrum, 2015). Therefore, we use RDKit to provide approximate samples from $p_\theta(L)$, and focus on developing a diffusion-based generative model to learn the distribution $p_\theta(\tau \mid L)$ over torsion angles, conditioned on the given graph and local structure.

**Conformer Matching** In (Jing et al., 2022), training on synthetic conformers produced by conformer matching has shown significant better performance than using ground truth alone. The conformer matching procedure operates as follows. For a molecule with $K$ conformers, it first generates $K$ random local structure estimates $\hat{L}$ using RDKit (Riniker & Landrum, 2015). To align these estimates with the ground truth conformers $C$, we compute a $K \times K$ cost matrix, where each entry represents the lowest RMSD achievable by adjusting the torsion angles of $\hat{L}$ to match $C$. We then solve the linear sum assignment problem on this approximate cost matrix to find the optimal matching between the true conformers $C$ and the estimates $\hat{C}$ (Crouse, 2016; Stärk et al., 2022). For each matched pair, it refines the alignment by performing a differential evolution optimization over the torsion angles to obtain the optimal conformer $\hat{C}$ (Méndez-Lucio et al., 2021). This complete assignment ensures consistency between the local structures seen during training and inference, preventing any distributional shift.

## D FURTHER RESULTS

In this section, we present additional experimental results to further validate the robustness and generalizability of our proposed fragment-based molecular modeling approach. We explore the performance of our method across different datasets, fragment size choices, and fragmentation strategies. Specifically, we provide detailed analysis on the following aspects: the results on both the GEOM-QM9 dataset, which consists of small molecules (with an average of 11 atoms per molecule), and the GEOM-XL dataset, which contains significantly larger molecules, averaging 132 atoms per molecule. For comparison, the GEOM-DRUGS dataset contains molecules with an average of 44 atoms. the impact of varying the fragment size control parameter, where only fragments larger than a specified threshold $z$ are selected during data augmentation; and the performance of chemical rule-based fragmentation under different data availability conditions, highlighting the method's adaptability to varying dataset sizes.

### D.1 FURTHER FRAGMENTATION STATISTICS

Figure 5 illustrates how the minimum fragment size $z$ affects the average number of fragments per molecule for the Graph-based, BRICS, and RECAP fragmentation methods across the GEOM-QM9, GEOM-DRUGS, and GEOM-XL datasets. GEOM-QM9, comprising small molecules averaging 11 atoms, shows that the Graph-based method generates significantly more fragments when $z$ is

---

**Algorithm 1** Training and Inference Procedure with Augmentation Phase

---

**Input:** Molecules $\{\mathcal{G}_0, \ldots, \mathcal{G}_N\}$ with ground truth conformations $\{C_{\mathcal{G},1}, \ldots, C_{\mathcal{G}_N,K}\}$; learning rate $\alpha$; number of conformations $K$; number of diffusion steps $\mathcal{S}$; maximum number of selected edges $\kappa$; minimum fragment size $z$.
**Output:** Trained score model $\mathbf{s}_\theta$; predicted conformations $\{C_1, \ldots, C_K\}$.

 1: **Augmentation Phase:**
 2: **for** each molecule $\mathcal{G}$ in $\{\mathcal{G}_0, \ldots, \mathcal{G}_N\}$ **do**
 3:      Identify all cut-edges in $\mathcal{G}$.
 4:      Let $b$ be the total number of cut-edges.
 5:      Randomly select $K = \min(b, \kappa)$ edges.
 6:      Decompose $\mathcal{G}$ into fragments by removing the selected edges.
 7:      Discard fragments smaller than $z$ atoms.
 8:      Add the remaining fragments to the augmented training set $\mathcal{F}$.
 9: **end for**
10: **Training Phase:**
11: **for** each fragment $\mathcal{G}$ in augmented training set $\mathcal{F}$ **do**
12:      **for** each ground truth conformation $C_{\mathcal{G},k}$ of $\mathcal{G}$ **do**
13:          Extract torsion angles $\boldsymbol{\tau}_{\mathcal{G},k}$ from $C_{\mathcal{G},k}$.
14:      **end for**
15: **end for**
16: **for** epoch $\leftarrow 1$ to epoch$_{\max}$ **do**
17:      **for** each fragment $\mathcal{G}$ in $\mathcal{F}$ **do**
18:          Sample $t \sim \text{Uniform}[0, 1]$.
19:          Randomly select a ground truth torsion angle set $\boldsymbol{\tau}$ from $\{\boldsymbol{\tau}_{\mathcal{G},1}, \ldots, \boldsymbol{\tau}_{\mathcal{G},K}\}$.
20:          Sample noise $\epsilon \sim p_t(\epsilon \mid \mathbf{0})$, where $p_t$ is a wrapped normal distribution with variance $\sigma^2(t) = \sigma_{\min}^{1-t} \sigma_{\max}^t$.
21:          Obtain noisy torsion angles: $\tilde{\boldsymbol{\tau}} = \boldsymbol{\tau} + \epsilon$.
22:          Construct noisy conformation $\tilde{C}_{\mathcal{G}}$ by applying $\tilde{\boldsymbol{\tau}}$ to $\mathcal{G}$.
23:          Predict torsion updates $\Delta\boldsymbol{\tau} = \mathbf{s}_\theta(\tilde{C}_{\mathcal{G}}, t)$.
24:          Compute loss: $\mathcal{L} = \left\| \Delta\boldsymbol{\tau} - \nabla_{\tilde{\boldsymbol{\tau}}} \log p_{t|0}(\tilde{\boldsymbol{\tau}} \mid \boldsymbol{\tau}) \right\|^2$.
25:          Update model parameters: $\theta \leftarrow \theta - \alpha \nabla_\theta \mathcal{L}$.
26:      **end for**
27: **end for**
28: **Inference Phase:**
29: **for** each molecular graph $\mathcal{G}$ **do**
30:      Initialize torsion angles $\boldsymbol{\tau}^T \sim p_T(\boldsymbol{\tau})$ (e.g., uniform over $[0, 2\pi]^m$).
31:      Construct initial conformation $C^T$ by applying $\boldsymbol{\tau}^T$ to $\mathcal{G}$.
32:      **for** $n \leftarrow \mathcal{S}$ down to 1 **do**
33:          Compute $t = n/\mathcal{S}$.
34:          Predict torsion updates $\Delta\boldsymbol{\tau} = \mathbf{s}_\theta(C^n, t)$.
35:          Sample noise $\boldsymbol{z} \sim \text{WrappedNormal}(\mathbf{0}, \mathbf{I})$.
36:          Compute step size $g(t) = \sigma_{\min}^{1-t} \sigma_{\max}^t \sqrt{2\ln(\sigma_{\max}/\sigma_{\min})}$.
37:          Update torsion angles:

$$\boldsymbol{\tau}^{n-1} = \boldsymbol{\tau}^n + \left( \frac{g^2(t)}{N} \Delta\boldsymbol{\tau} + \mathbf{g}(t)\boldsymbol{z} \right).$$

38:          Construct updated conformation $C^{n-1}$ by applying $\boldsymbol{\tau}^{n-1}$.
39:      **end for**
40:      Store the final conformation $C^0$.
41: **end for**

---

small, but the fragment count drops rapidly as $z$ increases due to the limited molecular size. BRICS and RECAP produce fewer fragments with less sensitivity to $z$ changes. In the GEOM-DRUGS dataset, with molecules averaging 44 atoms, all methods produce more fragments, but the Graph-based method still leads, and the decline in fragment numbers with increasing $z$ is more gradual. For the GEOM-XL dataset, containing large molecules averaging 132 atoms, all methods generate

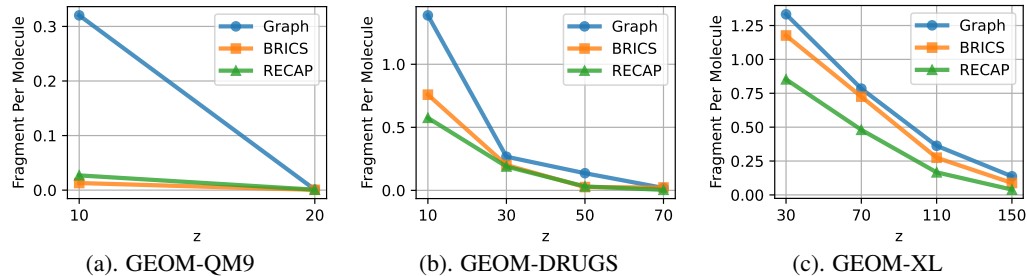

Figure 5: Average Fragment Number *v.s.* Minimum Fragment Size $z$ on different fragmentation methods.

Table 5: Quality of generated conformer ensembles for the GEOM-QM9 test set in terms of Coverage (%) and Average Minimum RMSD (Å) with $\delta = 0.5$ Å.

| Models | COV-R (%) ↑ | | AMR-R (Å) ↓ | | COV-P (%) ↑ | | AMR-P (Å) ↓ | |
| --- | --- | --- | --- | --- | --- | --- | --- | --- |
| | Mean | Median | Mean | Median | Mean | Median | Mean | Median |
| CGCF | 78.0 | 82.4 | 0.421 | 0.390 | 36.5 | 33.6 | 0.662 | 0.643 |
| ConFVAE | 77.8 | 88.2 | 0.415 | 0.373 | 38.0 | 34.7 | 0.622 | 0.609 |
| ConFGF | 88.4 | 94.3 | 0.267 | 0.268 | 46.4 | 43.4 | 0.522 | 0.512 |
| GeoDiff | 90.1 | 93.4 | 0.209 | 0.198 | 52.8 | 50.3 | 0.445 | 0.427 |
| RDKit | 85.1 | **100.0** | 0.235 | 0.199 | 86.8 | **100.0** | 0.232 | 0.205 |
| OMEGA | 85.5 | **100.0** | 0.177 | **0.126** | 82.9 | **100.0** | 0.224 | **0.186** |
| GeoMol | 91.5 | **100.0** | 0.225 | 0.193 | 86.7 | **100.0** | 0.270 | 0.241 |
| TorDiff | **92.8** | **100.0** | 0.178 | 0.147 | **92.7** | **100.0** | **0.221** | 0.195 |
| FADiff | **93.2** | **100.0** | **0.175** | 0.139 | **93.1** | **100.0** | **0.218** | 0.189 |

a higher number of fragments, and the differences between methods become less pronounced as $z$ increases. The Graph-based method remains the most sensitive to changes in $z$, while BRICS and RECAP display steady decreases. Overall, these trends highlight that larger molecules permit more fragmentation, and the Graph-based method consistently yields more fragments, especially at smaller $z$ values, whereas BRICS and RECAP are less influenced by the minimum fragment size due to their inherent fragmentation rules.

## D.2 FURTHER RESULTS ON GEOM-QM9 AND GEOM-XL

**Further Results on GEOM-QM9**  Table 5 presents the performance of various models on the GEOM-QM9 test set, which primarily consists of small molecules, making it a suitable benchmark for evaluating the ability of models to generate accurate conformer ensembles for relatively simple molecular structures. The results are evaluated with a threshold of $\delta = 0.5$ Å. Our proposed model, FADiff, achieves the highest overall performance, with a mean COV-R of 93.2% and a median of 100.0%, surpassing all other models. In terms of AMR-R, FADiff also outperforms the rest, with the lowest mean RMSD of 0.175Åand a competitive median of 0.139Å. For predicted conformers, FADiff maintains its superior performance, achieving a mean COV-P of 93.1% and a median of 100.0%, while also recording the lowest mean AMR-P of 0.218Åand a median of 0.189Å. These results demonstrate that FADiff not only generates highly accurate conformer ensembles but also ensures excellent coverage, outperforming other state-of-the-art methods such as TorDiff, GeoMol, and GeoDiff, which also show strong performance but fall short in both coverage and RMSD metrics. The performance of FADiff on GEOM-QM9 dataset highlights its effectiveness in capturing the torsional flexibility and geometric accuracy of simpler molecular structures.

**Further Visualizations on Conformation Generation for Large Molecules (GEOM-XL)**  Table 6 provides additional visualizations of conformer generation results on the GEOM-XL dataset, focusing on large molecules. These examples complement our earlier discussion on the superior generalization performance of FADiff on large molecules. The table compares the generated conformers from FADiff and TorDiff with the reference structures. It shows that FADiff produces

Table 6: Visualizations on conformation generation examples on GEOM-XL.

| Graph | Reference | FADiff | TorDiff |
| --- | --- | --- | --- |

conformers that closely resemble the reference structures on given examples, further elucidates the performance improvements presented in Table 2, highlighting FADiff's exceptional ability to handle large and complex molecules by generating conformers that closely match the reference structures.

### D.3 FURTHER RESULTS ON BOLTZMANN GENERATION

We follow the experimental setup from Jing et al. (2022) to evaluate FADiff, samples from the Boltzmann distribution over torsion angles. The evaluation is conducted on GEOM-DRUGS molecules We compare FADiff against TorDiff Jing et al. (2022) and annealed importance sampling (AIS) Neal (2001). The comparison focuses on the effective sample size (ESS) of 32 samples per molecule, which quantifies how closely the generated samples match the true Boltzmann distribution. ESS is computed using importance sampling weights, and performance is assessed across different temperatures to evaluate the sampling efficiency (Jing et al., 2022).

### D.4 FURTHER RESULTS ON CONFORMER MATCHING ABLATION

Table 7 presents an ablation study analyzing the impact of Conformer Matching during training on the GEOM-DRUGS test set. It reveals several key observations regarding the impact of Conformer Matching during training on the GEOM-DRUGS test set. Generally, models trained with CM outperform those without it, achieving higher Coverage percentages (COV-R and COV-P) and lower Average Minimum RMSD values (AMR-R and AMR-P).

However, an intriguing phenomenon occurs with TorDiff at $\mathbf{n} = 1000$: training *without* CM yields better performance than training *with* CM (mean COV-R of 45.39% vs. 34.60%, and mean AMR-R of 0.8190Å *v.s.* 0.8933Å). This suggests that, with limited data, directly using actual conformer structures for training may enhance TorDiff's generalization ability more than CM. As the dataset size increases, this advantage diminishes, and models trained without CM exhibit declining performance. This inverse relationship indicates that training on actual conformer data without CM may lead to overfitting to specific conformations, hampering generalization to unseen data as the model becomes more specialized on the training set. Conversely, CM helps prevent distributional shifts by

Table 7: Training with and without conformer matching (CM) on the GEOM-DRUGS test set in terms of Coverage (%) and Average Minimum RMSD (Å) with $\delta = 0.75$ Å.

| Samples | Models | COV-R (%) ↑ | | AMR-R (Å) ↓ | | COV-P (%) ↑ | | AMR-P (Å) ↓ | |
|---|---|---|---|---|---|---|---|---|---|
| | | Mean | Median | Mean | Median | Mean | Median | Mean | Median |
| n = 1000 | FADiff | **49.39** | **46.66** | **0.7928** | **0.7844** | **33.84** | **21.23** | **1.0455** | **0.9823** |
| | w/o CM | 47.68 | 45.45 | 0.8101 | 0.7861 | 33.39 | 20.93 | 1.0526 | 1.0056 |
| | TorDiff | 34.60 | 17.23 | 0.8933 | 0.8909 | 20.84 | 5.56 | 1.1897 | 1.1795 |
| | w/o CM | 45.39 | 39.44 | 0.8190 | 0.8040 | 28.74 | 15.00 | 1.1033 | 1.0667 |
| n = 10000 | FADiff | **62.82** | **69.70** | **0.6736** | **0.6505** | **43.10** | **37.72** | **0.9081** | **0.8840** |
| | w/o CM | 43.95 | 36.43 | 0.8353 | 0.8224 | 29.18 | 14.22 | 1.1038 | 1.0426 |
| | TorDiff | 52.76 | 54.10 | 0.7507 | 0.7379 | 33.88 | 20.64 | 1.0458 | 1.0371 |
| | w/o CM | 43.15 | 36.36 | 0.8478 | 0.8291 | 28.11 | 13.06 | 1.1051 | 1.0676 |
| Full | FADiff | **70.07** | **78.35** | **0.6092** | **0.5876** | **52.87** | **54.17** | **0.8003** | **0.7486** |
| | w/o CM | 37.52 | 25.00 | 0.8866 | 0.8863 | 23.73 | 8.22 | 1.1598 | 1.1307 |
| | TorDiff | 67.49 | 75.81 | 0.6339 | 0.6178 | 49.53 | 47.16 | 0.8269 | 0.7782 |
| | w/o CM | 34.99 | 20.88 | 0.9326 | 0.9174 | 23.08 | 8.13 | 1.1803 | 1.1340 |

Table 8: Results of Property Prediction task.

| Method | $E$ | $E_{\min}$ | $\Delta\epsilon$ | $\Delta\epsilon_{max}$ |
|---|---|---|---|---|
| RDKit | 0.92 | 0.65 | 0.37 | 0.80 |
| GeoMol | 0.38 | 0.19 | 0.29 | 0.81 |
| GeoDiff | 0.26 | **0.13** | 0.31 | 0.70 |
| TorDiff | 0.20 | 0.14 | 0.23 | **0.43** |
| FADiff | **0.19** | **0.13** | **0.20** | **0.43** |

aligning training and inference conformer distributions, which becomes increasingly beneficial with larger datasets. FADiff consistently outperforms TorDiff when CM is applied, suggesting it more effectively leverages CM for improved conformer generation. Overall, incorporating CM during training enhances model performance and generalization, especially with larger datasets, whereas training without CM may offer short-term benefits with very limited data but ultimately hinders performance as data volume grows.

### D.5 FURTHER EXPERIMENTAL RESULTS ON PROPERTY PREDICTION TASK

We adopt the property prediction task setup from (Xu et al., 2022; Shi et al., 2021), where 30 molecules from the GEOM-DRUGS dataset are used, with 50 samples generated for each molecule. The PSI4 toolkit is employed to compute the energy ($E$) and HOMO-LUMO gap ($\epsilon$) for each conformer, and comparisons are made with the ground truth for average energy ($E$), minimum energy ($E_{\min}$), average gap ($\Delta\epsilon$), and maximum gap ($\Delta\epsilon_{max}$). As shown in Table 8, our method generates the most chemically accurate ensembles.

## E  LIMITATIONS AND FUTURE WORK

While our fragment-based augmentation approach has demonstrated significant improvements in generating accurate and diverse molecular conformations, there are several limitations that present opportunities for future research. First, when applying fragmentation methods as a general data augmentation technique for data-driven computational models, we may encounter unmanageable data volumes, especially when training with large molecular datasets and setting low fragment size thresholds, as discussed in our appendix on fragmentation statistics. This highlights the need for more data-efficient frameworks. Leveraging prior domain knowledge, such as scaffold networks or molecular graphs (Quinn et al., 2017; Nothias et al., 2020; Kruger et al., 2020), could enhance data efficiency during the fragmentation process, reducing the computational burden while preserving essential chemical information.

From a methodological standpoint, while fragment-based augmentation has delivered impressive results within the torsional diffusion framework, there is room for further improvement to fully capitalize on the benefits of data augmentation. The current framework relies on cutting edges in graph structure algorithms, which introduces limitations—such as difficulty in modeling fully

connected supramolecular structures where rotating edges alone cannot capture reasonable conformations. Potential solutions include introducing additional variations in Euclidean space, like incorporating ring-connecting edges from junction trees and allowing non-rigid rotational edges that permit changes in relative atomic distances (Jin et al., 2018). Additionally, integrating bond stretching and angle bending components into conformational energy modeling could address challenges in representing fully connected structures, effectively combining elements of methods like GeoDiff with Torsional Diffusion (Jing et al., 2022; Xu et al., 2022). By exploring the chemical underpinnings of fragment effectiveness, we can gain deeper insights that enable the development of more effective chemical modeling processes, reduce errors, and enhance data learning efficiency through interdisciplinary collaboration. Moreover, our method has the potential to significantly advance computation-driven approaches by greatly increasing the amount of available data, which is crucial for the success of machine learning models. Scaling laws indicate that enhancing model capacity and expanding training datasets with a robust foundational framework can markedly improve performance (Frey et al., 2023). By utilizing our fragmentation approach to augment data and scaling up model parameters, we open new avenues for designing and training larger computational models in physical chemistry, potentially unlocking novel applications in chemical and materials science (von Lilienfeld et al., 2020; Sadybekov & Katritch, 2023).

