# OpenReview forum: "Fragment-Augmented Diffusion for Molecular Conformation Generation"
_ICLR.cc/2025/Conference — Submitted to ICLR 2025_

### Official Review · Reviewer_eJ6N · 2024-11-03

**Soundness:** 2
**Presentation:** 3
**Contribution:** 2
**Rating:** 3
**Confidence:** 3

**Summary:**

This work focuses on the molecular conformer generation task, where the model is trained to generate the 3D structures given the chemical graph. This work closely follows the previous Torsional Diffusion work (Jing et al., 2022), which considers performing diffusion on torsion angles instead of atom coordinates to reduce the degree of freedom.

With the context, the main contribution of this work is to propose to use the molecular fragmentation as augmented data for the Torsional Diffusion training (Fragment-Augmented Diffusion, i.e., FADiff). Specifically, given a molecule in the training set, FADiff first decomposes the molecule into multiple fragments and then treats each fragment (that satisfies some criterias) as separate training data. The 3D structures of these augmented fragments are “cropped” from the 3D structures of the original molecule where the fragments come from. The experiments on conformer generation shows that FADiff can outperform the previous Torsional Diffusion work.

**Strengths:**

- Exploring to use fragments as a data augmentation strategy is interesting.
- The manuscript is well written and organized. It is easy to read and follow.
- It is interesting to see FADiff achieves good results in a data-scarce setting, which demonstrates its effectiveness.

**Weaknesses:**

The idea of using fragments as a data augmentation strategy is interesting but not well justified. Several questions remain unanswered.
- The torsion angles in a molecule are determined by the global structure rather than its local structure in the decomposed fragment. Why augmenting the training with local 3D structures cropped from the global structures can help the performance, given that all the local structure information is contained in the global structure?
- How can we demonstrate that if the performance gain indeed comes from the data augmentation or it is because the model architecture in the original Torsional Diffusion constrains the learning from the local structure?
- The REAL structures for fragments are unknown. By using the structures cropped from the global structures as the local structures for training, would this mislead the model on learning the underlying physics?

**Questions:**

- Do you use the original molecule for training in addition to the fragments?
- It is mentioned that the experimental setup follows Jing et al., 2022. The results shown in Table 1 for TorDiff are different from (worse than) the original results reported by authors. Where does such difference come from?

---

### Official Review · Reviewer_tQne · 2024-11-03

**Soundness:** 2
**Presentation:** 3
**Contribution:** 2
**Rating:** 5
**Confidence:** 3

**Summary:**

Through this paper, the authors propose Fragment-Augmented Diffusion (FADiff), a framework that integrates molecular fragmentations into diffusion models as a data augmentation strategy to enhance molecular conformation generation.

**Strengths:**

- Overall, the paper was easy to follow. The writing and the concept figure were clear.
- The authors provided the codebase.
- Error analysis regarding the choice of decomposition edges was included.

**Weaknesses:**

Weaknesses
I will combine the *Weaknesses* section and the *Questions* section. My concerns are as follows:
- The reported results of TorDiff in Table 1 are significantly worse than those in the original paper [1]. Compared to the results in the original paper, FADiff is worse than TorDiff. Can you provide an explanation for this?
- There is no analysis on runtime. I strongly suggest to compare time efficiency of the proposed methods with baselines (e.g., Table 2 in [1]).
- TorDiff provided the errors of E, ∆ε, Emin, and μ for the generated ensembles. It would be great to provide the same comparison in this paper, as it would allow for a more robust comparison.
- As the authors acknowledged, the fragmentation method should be carefully chosen under the proposed framework to ensure the preservation of key chemical properties.
- On line 96, the term molecular generation was used, which is different from molecular conformation generation.

---

**References:**

[1] Jing et al., Torsional diffusion for molecular conformer generation, NeurIPS, 2022.

**Questions:**

Please see the *Weaknesses* section for my main concerns.

For now, I’m leaning toward borderline reject, but I’ll be glad to raise the score when all the questions are fully addressed.

---

### Official Review · Reviewer_U1y1 · 2024-11-04

**Soundness:** 2
**Presentation:** 3
**Contribution:** 2
**Rating:** 5
**Confidence:** 4

**Summary:**

This paper proposes to improve diffusion model’s generalizability by augmenting training datasets with a wide range of fragment-level configurations. The fragments used for augmentations are selected to be chemically valid, while preserving essential properties of the original molecule. The authors provide justification for approximating the labels for fragments with the concept of mutual information. Given the augmented dataset, FADiff outperforms previous baseline models for molecule conformer generation. The model architecture and training scheme itself follows that of Torsional Diffusion, and has no novelty.

**Strengths:**

- The motivation of applying fragmentation methods for data augmentation is chemically well-constructed. The understanding of torsional angles in fragments and is insightful, demonstrating promising applications in future works.
- The paper outperforms selected baseline models in two of the three benchmark datasets.
- Paper is well-written and comprehensible. The code and dataset is available, enabling reproducibility.

**Weaknesses:**

- The main task required for this paper is to demonstrate the usefulness of fragment-based augmentation. Nonetheless, the validity of using the selected fragmentation methods to formulate the augmented datasets is not sufficient.
- Clarification upon the augmentation procedure, particularly regarding the processing of redundant fragments, is required.
- More recent baseline models should be compared. Three recently published generative models are listed in the Major questions section.

**Questions:**

Major Questions
- According to the described methodology, identical/highly similar fragments generated from different molecules would have distinct torsion angles, i.e., same input fragments will have different output labels. How are the conflicting representations in such cases handled?
- The two fragmentation methods (RECAP, BRICS) cover similar chemical deconstruction. In cases where the two fragmentation methods yield overlapping fragments within the same molecule, how is it processed? How did the authors deal with potential data bias for frequently observed fragments?
- Selecting the right fragmentation method is indeed important, but the authors fail to sufficiently explain how the selected fragmentation methods optimal considering error variance. Provide quantitative elaboration on error analysis according to fragmentation methods.
- The model lacks comparison with more recent generative models for molecule conformer generation. The following models should be added as baseline models.
1. DiSCO: Diffusion Schrodinger Bridge for Molecular Conformer Optimization (Lee et al., 2024)
2. Equivariant Flow Matchiing for Molecular Conformer Generation (Hassan et al., 2024)
3. Swallowing the Bitter Pill: Simplified Scalable Conformer Generation (Wang et al., 2024)


Minor Questions
- Provide statistics of the augmented datasets on components (e.g. number of fragments, average length, average fragment generated from each molecule).
- Can this data augmentation method be model agnostic and be applied to other generative models as well?
- The main results are evaluated upon the train/val/test splits setup from TorDiff but significant variation is present in the result of GEOM-DRUGS. Why is the dataset not reproducible unlike other datasets (GEOM-QM9, XL)?
- Appendix Figure 2 requires more explanation and connection to the main text.
- Authors additionally used conformer matching method, to align ground truth conformers with synthetic RDKit generated conformers. Also validate the effect of this data augmentation technique.

---

### Meta-Review · Area_Chair_JwNC · 2024-12-14

**Metareview:**

The submission provides a method to enhance diffusion-based organic molecular structure generation, by cropping fragments out of the original molecular structures as data augmentation. Although the idea is worth investigating in general, the finish does not seem solid enough for the current version. I agree with the reviewers that if following Torsional Diffusion to model molecular structures, the torsional angles are not sufficiently local and may not be determined by a fragment alone. Moreover, the paper does not seem to have solidly ablated the effect of using fragment augmentation and there are doubts on the baseline results. The authors did not respond to these concerns. I hence recommend a reject.

**Additional Comments On Reviewer Discussion:**

The authors did not respond to the reviewers, and the reviewers did not update their scores.

---

### Decision · Program_Chairs · 2025-01-22

Reject